# Genetically personalised organ-specific metabolic models in health and disease

Carles Foguet [1,2,3,4] ✉, Yu Xu [1,3,4], Scott C. Ritchie [1,3,4,5,6], Samuel A. Lambert [1,2,3,4], Elodie Persyn[1,3,4], Artika P. Nath[1,6], Emma E. Davenport[7], David J. Roberts[8,9,10], Dirk S. Paul [3,4,5], Emanuele Di Angelantonio[2,3,4,5,9,11], John Danesh[2,3,4,5,7,9], Adam S. Butterworth [2,3,4,5,9], Christopher Yau[12,13] & Michael Inouye [1,2,3,4,5,6,14] ✉

Understanding how genetic variants influence disease risk and complex traits (variant-to-function) is one of the major challenges in human genetics. Here we present a model-driven framework to leverage human genome-scale metabolic networks to define how genetic variants affect biochemical reaction fluxes across major human tissues, including skeletal muscle, adipose, liver, brain and heart. As proof of concept, we build personalised organ-specific metabolic flux models for 524,615 individuals of the INTERVAL and UK Biobank cohorts and perform a fluxome-wide association study (FWAS) to identify 4312 associations between personalised flux values and the concentration of metabolites in blood. Furthermore, we apply FWAS to identify 92 metabolic fluxes associated with the risk of developing coronary artery disease, many of which are linked to processes previously described to play in role in the disease. Our work demonstrates that genetically personalised metabolic models can elucidate the downstream effects of genetic variants on biochemical reactions involved in common human diseases.

Genome-wide association studies (GWAS) have identified more than 50,000 genetic variants associated with complex traits or diseases[1]. While the contribution of individual variants to a given phenotype is generally small, the effect of multiple genetic variants can be aggregated into polygenic scores (PGS), which are highly predictive of disease incidence and enhance existing risk models[2–4]. However, while GWAS and PGS can be useful for risk stratification[5–7], the mechanisms

by which genetic variants influence disease risk, i.e., variant to function (V2F), remain largely unsolved. Addressing V2F is a major challenge in human genetics and has the potential to unveil many new therapeutic targets[6,8,9].

An approach to address the V2F challenge is to quantify how genetic variation causes disease through the regulation of molecular traits. To this end, genetic variants affecting gene expression are

[1]Cambridge Baker Systems Genomics Initiative, Department of Public Health and Primary Care, University of Cambridge, Cambridge, UK. [2]Health Data Research UK Cambridge, Wellcome Genome Campus and University of Cambridge, Cambridge, UK. [3]British Heart Foundation Cardiovascular Epidemiology Unit, Department of Public Health and Primary Care, University of Cambridge, Cambridge, UK. [4]Heart and Lung Research Institute, University of Cambridge, Cambridge, UK. [5]British Heart Foundation Centre of Research Excellence, University of Cambridge, Cambridge, UK. [6]Cambridge Baker Systems Genomics Initiative, Baker Heart and Diabetes Institute, Melbourne, VIC, Australia. [7]Wellcome Sanger Institute, Hinxton, UK. [8]BRC Haematology Theme, Radcliffe Department of Medicine, and NHSBT-Oxford, John Radcliffe Hospital, Oxford, UK. [9]National Institute for Health and Care Research Blood and Transplant Research Unit in Donor Health and Behaviour, University of Cambridge, Cambridge, UK. [10]NHS Blood and Transplant, John Radcliffe Hospital, Oxford, UK. [11]Health Data Science Centre, Human Technopole, Milan, Italy. [12]Nuffield Department of Women's and Reproductive Health, University of Oxford, Oxford OX3 9DU, UK. [13]Health Data Research UK, Gibbs Building, 215 Euston Road, London NW1 2BE, UK. [14]The Alan Turing Institute, London, UK. ✉ e-mail: cf545@medschl.cam.ac.uk; mi336@medschl.cam.ac.uk

identified and subsequently aggregated into models that can impute the abundance of transcripts and proteins[10–12]. For example, PredictDB is a database that offers a collection of linear models to impute transcript levels in specific organs of the human body[13]. PredictDB models were trained in the GTEx dataset, which contains genotype profiling and tissue-specific transcript abundance from post-mortem donors[14]. Imputed transcript or protein levels can be used to perform transcriptome-wide or proteome-wide association analyses, respectively, to identify gene products associated with disease[10,15]. Alternatively, PGSs for disease can be used to identify proteins and other gene products which may disrupt polygenic risk[16]. However, transcripts and proteins do not exert their effects in isolation but in highly connected and complex biological networks. Indeed, previous studies have shown the merit of analysing genetic variation in the context of gene co-expression and gene interaction networks to characterise how the effects of genetic variants contribute to complex traits or diseases by propagating through biological networks[17–20].

Metabolism is one of the most prominent biological networks and a comparatively tractable experimental setting in which to address the V2F challenge. Essentially, metabolism is a set of interconnected chemical reactions and transport processes occurring in a highly ordered, regulated and coordinated manner across multiple organs in the human body[21]. The metabolic phenotype of a given organ is defined by both metabolite concentrations and metabolic fluxes (i.e., the rates at which substrates are converted to products through reactions) and emerges from the complex interaction of metabolites, enzymes, and transmembrane carriers[22,23]. Metabolite concentrations offer a static snapshot of metabolite distributions, whereas metabolic fluxes provide a map of metabolite traffic through metabolic pathways[24].

Genome-scale metabolic models (GSMMs), mathematical representations of the metabolic reaction network arising from the human genome[25,26], simulate steady-state metabolic fluxes by formulating network stoichiometry as sets of linear equations and directionality constraints[27]. GSMMs have emerged as a useful approach to integrate transcriptomics, proteomics, and metabolomics to characterise metabolic flux maps[28,29]. For example, proteomics, metabolomics, and physiological data have been used to build human organ-specific GSMMs[30]. Similarly, there is increasing interest in integrating individual measures to build personalised GSMMs that reflect the specific metabolic phenotype in each individual, thus facilitating personalised medicine[30–34].

Since gene expression is highly heritable[10,13], it may be feasible to leverage human genome-scale metabolic networks to analyse the system-wide effects of genetic variants on metabolism and build genetically personalised GSMMs. To this end, we present a framework where transcript levels imputed from genetic data can be used to simulate personalised and organ-specific, genome-scale flux maps using the quadratic metabolic transformation algorithm (qMTA). Such flux maps provide genetically personalised metabolic models at a genome scale for each tissue. As proof of concept, we build personalised organ-specific flux maps for over 520,000 individuals across the INTERVAL[35,36] and UK Biobank (UKB)[37] cohorts, then perform a fluxome-wide association study (FWAS) to test the association between organ-specific flux values and directly measured blood metabolite levels. Finally, we apply FWAS to identify fluxes associated with coronary artery disease (CAD), thus demonstrating the potential of genome-scale flux maps for V2F by elucidating intermediary biochemical reactions between genetic variation and common disease.

## Results

### A computational framework for genetically personalised organ-specific GSMMs

We developed a framework for building personalised organ-specific flux maps from genotype data (Fig. 1; Methods). First, we extract the organ-specific models from the Harvey/Harvetta multiorgan model[30], which provide a set of curated metabolic networks for the main organs of the human body. Harvey/Harvetta models were built from the Recon3D human GSMM[25], which has been superseded by HUMAN1[26]. HUMAN1 shares 97% of reactions with Recon3D, but it incorporates a myriad of improvements in gene-reaction rules, reaction reversibility and stoichiometric consistency compared to the latter. Hence, we performed a liftover of the Harvey/Harvetta organ-specific models to HUMAN1 (Methods).

With the HUMAN1-based organ-specific models, the next step is to compute a reference flux distribution for each organ under consideration. This is achieved by defining organ-specific metabolic objectives that must be fulfilled (e.g., synthesis of neurotransmitters in the brain), obtaining average organ transcript abundances from GTEx[14] and using them as an input for the GIM3E algorithm[38]. GIM3E is an algorithm that, subject to fulfilling the organ-specific metabolic objectives, seeks to minimise the overall flux through the network using transcript abundance data to give each reaction a minimisation weight inversely proportional to the expression of the enzymes catalysing it. Subsequently, flux sampling[39] is applied to identify a representative flux distribution (i.e., sets of flux values) in the solution space within 99% of the GIM3E optimal solution. The resulting set of flux values, termed reference flux distribution, is both enzymatically efficient and consistent with the average transcript abundances in each organ (Supplementary Fig. 1). The flux distribution can be assumed to represent the average metabolic state of each modelled organ in the general population.

Subsequently, models from PredictDB[13] are used to impute personalised organ-specific transcript abundances from genotype data. The resulting imputed transcript data are mapped to reactions in the organ-specific subnetworks as putative reaction activity fold changes relative to the average organ-specific transcript expression in GTEx[14]. The imputed personalised reaction activity fold changes and the reference flux distributions are then utilised by the qMTA to compute genetically personalised organ-specific flux maps. Briefly, qMTA finds the flux distributions most consistent with the putative reaction activity fold changes in each individual (Supplementary Fig. 2; Methods).

### Building flux maps for >520,000 individuals

Using the above framework, we built personalised organ-specific flux maps for 37,220 and 487,395 individuals from the INTERVAL[35,36] and UKB[37] cohorts, respectively. Personalised models were generated for skeletal muscle, adipose tissue, liver, brain, and heart, which together account for roughly 66% of body weight in an average adult[40]. Overall, 14,220 reaction flux values were computed for each individual. Metabolic fluxes "flow" through pathways where the product of one reaction is the substrate of successive reactions; thus, many of the flux values computed in each individual will have inherent dependencies (Supplementary Fig. 3A–C). As such, from the 14,220 reaction flux values, we selected a subset of 4300 flux values without strong pairwise correlations ($\rho < 0.9$) for further analysis (Supplementary Fig. 3D; Methods).

Principal component analysis of the personalised organ-specific flux values for individuals of INTERVAL and UKB showed the underlying structure in the data (Supplementary Fig. 4). Fluxes with the greatest loadings on top principal components (PCs) tended to be related to the known metabolism of each organ (Supplementary Data 1). For example, in the liver, fluxes through reactions and transport processes of amino acid, glycerophospholipid, and nucleotide metabolism exhibited large loadings along the first five PCs. Key reactions in cholesterol and bile acid biosynthesis also had large PC loadings, reflecting the function of the liver in cholesterol homoeostasis[21]. In both skeletal muscle and heart, the top PCs were associated with fluxes through transport processes of amino acids and reactions related to fatty acid β-oxidation, processes which play key roles in skeletal muscle and heart[41–44]. Notably, in the brain, the main

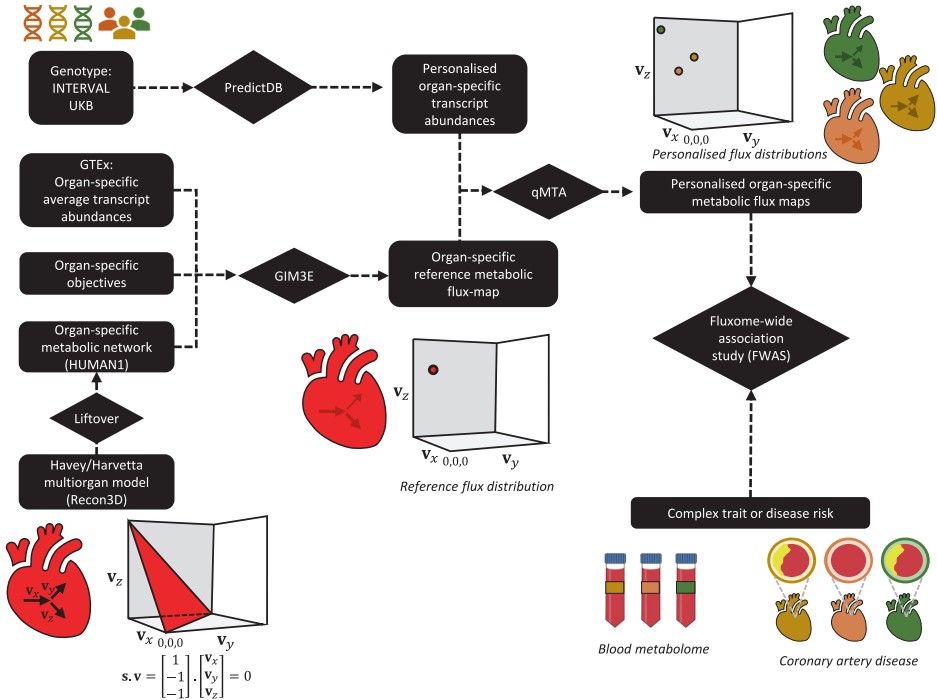

**Fig. 1 | Framework for computing organ-specific personalised genome-scale flux maps from genotype data and performing fluxome-wide association study (FWAS).** First, we extract the organ-subnetworks from the Harvey/Harvetta multi-organ models, which were built from Recon3D, and we perform a liftover to update them to HUMAN1, the most recent human GSMM. Then, a reference flux map is computed for each organ using the GIM3E algorithm to integrate average transcript abundances and organ-specific metabolic objectives into the organ-specific meta-bolic subnetwork. In parallel, personalised organ-specific transcript abundances are imputed from genotype data of the INTERVAL and UK Biobank (UKB) cohorts using the models from PredictDB. Next, the quadratic metabolic transformation algorithm (qMTA) is used to integrate the organ-specific transcript abundances and reference flux distribution and compute personalised organ-specific metabolic flux maps. The resulting flux maps can be used to perform FWAS to complex traits or diseases such as blood metabolic features or coronary artery disease. A hypothetic representation of an organ-specific solution space, reference flux distribution, and a set of three personalised flux distributions is shown for a reaction network with three fluxes ($\mathbf{v}_x$, $\mathbf{v}_y$ and $\mathbf{v}_z$).

loadings on the top principal components were distributed across a wide range of pathways. For instance, PC1 was associated with reactions and transport processes involving bile acids and their precursors. Bile acids, which can be synthesised within the brain and can also be transported across the blood-brain barrier, have been reported to act as regulators of neurological functions[45,46]. Likewise, PC2, and to a lesser extent PC3, were related to reactions and transport processes from amino acid metabolism, including reactions linked to neuro-transmitters such as dopamine, glycine, glutamate, and nitric oxide. PC4 was associated with reactions of fatty acid metabolism, most notably several reactions involving arachidonic acid, a conditionally essential fatty acid with many roles in brain function in health and disease[47,48]. Lastly, PC5 was primarily associated with reactions of nucleotide metabolism. Finally, in adipose tissue, all PCs were strongly associated with fatty acid metabolism, including reactions involved in their oxidation, biosynthesis and transport. However, PC2 and PC3 were also associated with reactions of steroid metabolism, reflecting adipose tissue's capacity to synthesise and convert steroids[49].

### Fluxome-wide association study for blood metabolites

We next validated that genetically personalised GSMMs could generate reliable and meaningful flux predictions across cohorts. As phenotypes, organ-specific flux maps are expected to lead to distinct profiles in the blood metabolome. To demonstrate this, we performed an association analysis by individually regressing each measured blood metabolic feature against the 4300 personalised fluxes computed in the INTERVAL[35,36] and UKB[37] cohorts (Supplementary Fig. 2; Methods). The blood metabolome for INTERVAL comprised both Nightingale Health NMR assays ($N$ = 37,720 participants) and Metabolon HD4 mass

spec assays ($N$ = 8115 participants)[50]. In UKB, blood samples for 120,266 participants were profiled with Nightingale Health NMR[51].

For INTERVAL, an FDR-adjusted significance threshold of $P < 1.0 \times 10^{-6}$ was applied to control for all tested pairs (Methods). We identified 4312 significant associations between flux values and blood metabolic features in total, of which 1066 were for the Nightingale platform and 3246 for Metabolon (Supplementary Data 2, Fig. 2A, B). Consistent with the role of the liver in whole-body metabolic homoeostasis[21], the liver was the organ with the most associations (1301), followed by the heart (1005), skeletal muscle (896), brain (593), and adipose tissue (517) (Fig. 2C, Supplementary Data 2). We externally validated the INTERVAL flux associations with Nightingale metabolites using UKB (Fig. 2D). We found 83% of the INTERVAL associations replicated in UKB with an FDR-adjusted significance threshold of $P < 1.0 \times 10^{-6}$ and consistent direction of the effect sizes. Effect sizes were themselves highly correlated ($\rho = 0.82$) between INTERVAL and UKB (Supplementary Fig. 5).

Finally, we also evaluated the effect of the underlying genome-scale reconstructions of human metabolism in the FWAS for blood metabolic features. With this aim, we used organ-specific models built from the Recon3D human GSMM[25,30] to compute genetically-personalised fluxes for the INTERVAL cohort[35,36], test their association to blood metabolic features, and compare the results to the above-described FWAS that had used fluxes computed with HUMAN1-based models. We identified 3895 significant associations between blood metabolic features and the genetically personalised flux values com-puted using Recon3D-based organ-specific models (Supplementary Data 2). There was a significant overlap with HUMAN1 models as 1761 of these associations could be replicated in the HUMAN1-based FWAS, and

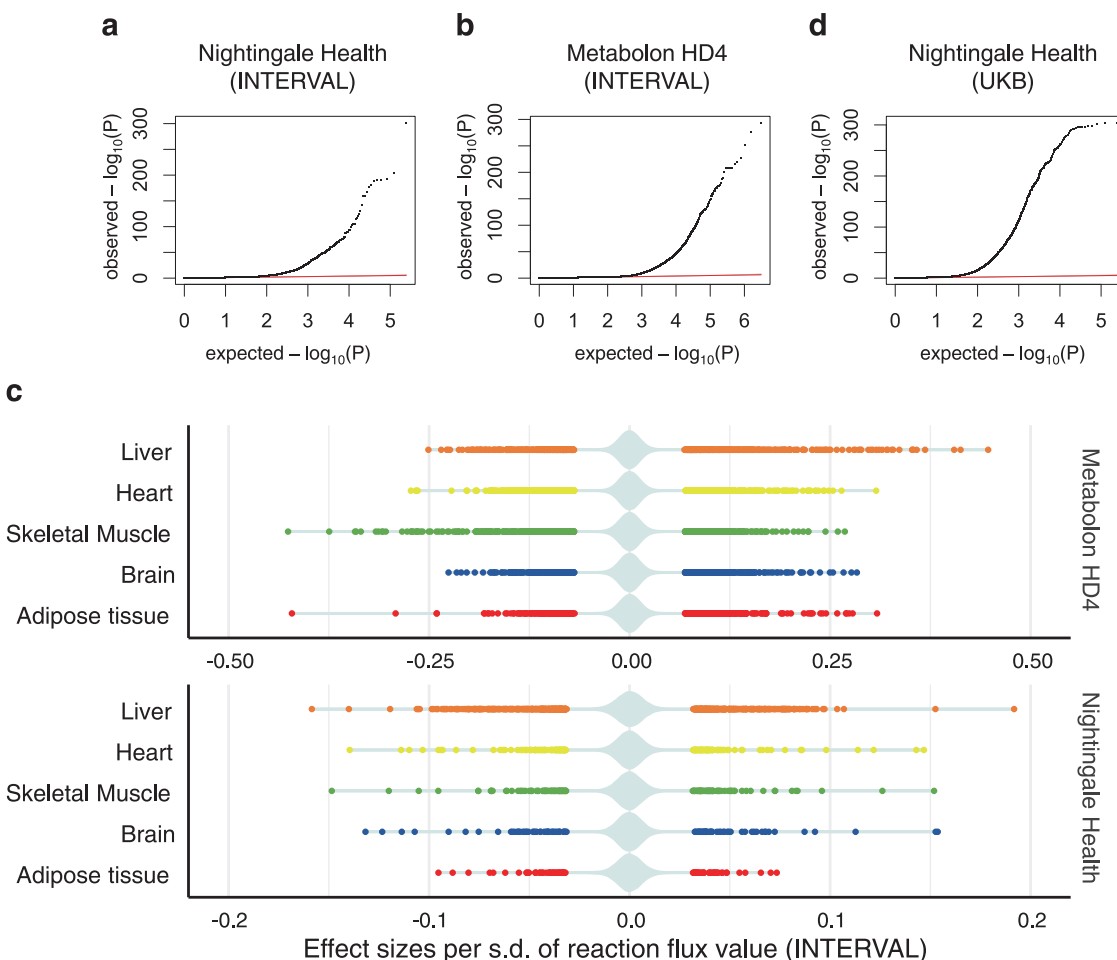

**Fig. 2 | Fluxome-wide association study (FWAS) between genetically personalised flux maps and blood metabolic features. a, b, d** Quantile-quantile (QQ) plot of the observed *P* values for associations between flux values and blood metabolic features measured with the Nightingale Health platform in INTERVAL (**a**), Metabolon HD4 platform in INTERVAL (**b**) and Nightingale Health platform in UK Biobank (**d**). The red lines indicate the expected distribution of *P* values under a uniform distribution (i.e., null hypothesis). **c** Plot of statistically significant (FDR-adjusted *P* value < 10⁻⁶) flux effect sizes per organ to blood metabolic features measured with either the Nightingale Health or Metabolon HD4 assay in INTERVAL. A violin plot, coloured in pale azure, shows the distribution of both significant and non-significant effect sizes. The statistical significance of each flux to blood metabolic feature associations was evaluated with linear regression (two-tailed *t*-test for flux effect size; Methods).

the associated effect sizes on blood metabolites were highly correlated between HUMAN1 and Recon3D analyses ($\rho = 0.72$). However, 2134 associations were only statistically significant on the Recon3D-based analysis and could not be replicated with HUMAN1 models. Likewise, of the 4312 significant associations between blood metabolic features and fluxes computed using HUMAN1 models, 2551 associations could not be detected with Recon3D-based models. Such discrepancy between HUMAN1- and Recon3D-based analyses is not surprising; HUMAN1[26], which is a newer reconstruction of human metabolism than Recon3D[25], expands gene reaction annotations and refines reaction reversibility, both of which can have significant effects on how genetic variation propagates through the network and, thus, can lead to significant differences in the resulting personalised flux maps and the downstream FWAS. Indeed, many discrepancies between the Recon3D and HUMAN1 results are likely artefacts emerging from erroneous or incomplete annotations in Recon3D. Throughout this work, we focus on the analyses and discussion of HUMAN1-based fluxes, as HUMAN1 has been established to be a better representation of human metabolism[26], but results obtained with Recon3D-based models will also be provided in the appropriate supplementary data (Supplementary Data 2, Supplementary Data 3, and Supplementary Data 4).

**Fluxome associations by metabolic feature class and reaction system**

The 4312 significant associations comprised 229 unique blood metabolic features and 763 unique organ-specific metabolic fluxes. Consistent with the coverage of the Nightingale Health and Metabolon HD4 platforms, we found that most of these blood metabolic features were lipid-related (Fig. 3A, Supplementary Data 2). Glycerides and phospholipids were enriched in associations across all organs relative to all features profiled with the Metabolon HD4 assay (Methods), suggesting an association with core reactions (i.e., active in all modelled organs). The liver and adipose tissue were also enriched in associations with steroids, reflecting the role of such organs in cholesterol[21] and steroid hormone metabolism[49].

We further assessed the metabolic systems of the 763 organ-specific metabolic fluxes from the significant associations (Fig. 3B, Supplementary Data 2) and found that most reactions were functionally part of lipid metabolism, consistent with a large number of associations with lipid metabolic features. Reactions of fatty acid metabolism were significantly enriched in associations with blood metabolic features in all organs relative to all analysed reactions in each organ-specific metabolic network. In the liver, reactions of

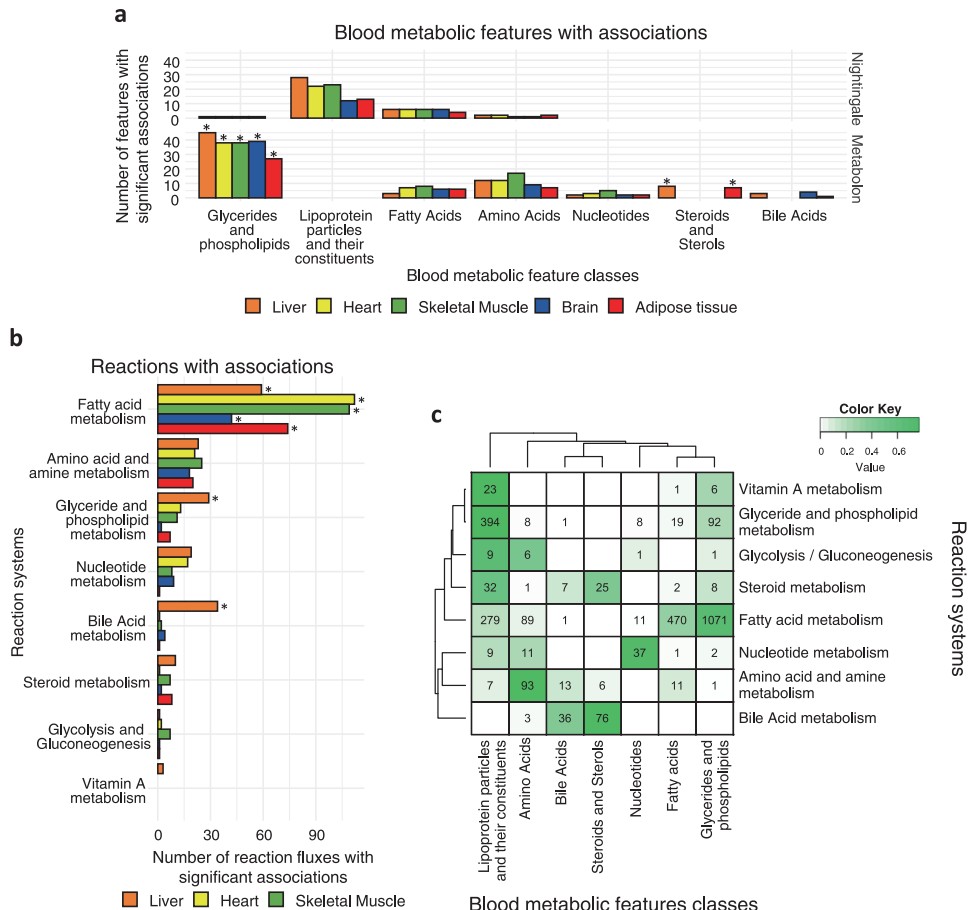

**Fig. 3 | Characterisation of the significant associations between blood metabolic features and metabolic fluxes.** **a** Classes of blood metabolic features with one or more significant associations to the fluxome. *denotes classes that are significantly enriched (one-sided Fisher's exact test, FDR-adjusted *P* value < 0.05). FDR-adjusted *P* values for significantly enriched classes: Glycerides and phospholipids (Metabolon): Liver: $2.5 \times 10^{-12}$, Brain: $3.7 \times 10^{-12}$, Heart: $6.9 \times 10^{-12}$, Skeletal Muscle: $2.9 \times 10^{-11}$, Adipose tissue: $3.3 \times 10^{-06}$; Steroids and sterols (Metabolon): Adipose tissue: 0.019, Liver: 0.045. Unannotated features and classes with few features are omitted for clarity. **b** Systems of the reactions whose flux values are significantly associated with one or more blood metabolic features. *denotes systems that are significantly enriched (one-sided Fisher's exact test, FDR-adjusted

*P* value < 0.05). FDR-adjusted *P* values for significantly enriched systems: Fatty acid metabolism: Skeletal muscle: $3.3 \times 10^{-30}$, Heart: $3.6 \times 10^{-27}$, Adipose tissue: $2.0 \times 10^{-17}$, Brain: $1.2 \times 10^{-15}$, Liver: $5.5 \times 10^{-10}$; Bile acid metabolism: Liver: $2.6 \times 10^{-05}$; Glyceride and phospholipid metabolism: Liver: 0.041. Unannotated reactions and systems with few features are omitted for clarity. **c** Heatmap of the intersection between blood metabolic feature classes and reaction systems in significant associations. Numbers at each intersection denote the number of significant associations between reaction fluxes of a given system and blood metabolic features of a given class. The colour key denotes the fraction of reactions of each system in each intersection.

glyceride and phospholipid metabolism and bile acid metabolism were also enriched.

There was widespread consistency between biochemical pathways and blood metabolic feature classes (Fig. 3C, Supplementary Data 2). For example, reactions from the glycerides and phospholipids system were primarily associated with blood metabolic features of glycerides and phospholipids as well as lipoprotein fractions and their constituents. We found that reactions of fatty acid metabolism were associated mainly with blood glyceride and phospholipids, followed by fatty acids, which themselves provide acyl chains to glycerides and phospholipids. Similarly, reactions from nucleotide metabolism and amino acid metabolism were primarily associated with blood metabolic features of nucleotides and amino acids, respectively.

### Fluxes of the hepatic triacylglycerol to cholesteryl ester pathway and blood lipoproteins

In the liver, we identified 555 associations between fluxes and lipoprotein fractions (Supplementary Data 2). Most of these associations were to reactions of glycerides and phospholipids metabolism, which were enriched in associations relative to all analysed liver fluxes

(Methods; Fig. 3B). FWAS revealed that a major determinant of triacylglycerols (TAG), free cholesterol (FC), and cholesteryl esters (CE) fractions in lipoproteins was a sequential set of reactions which we term the TAG to cholesterol esterification (TAG-CE) pathway (Fig. 4). In the TAG-CE pathway, TAGs are hydrolysed to diglycerides and fatty acids in the liver, diacylglycerides are then used as a substrate to synthesise phospholipids (i.e., phosphatidylcholine and phosphatidylethanolamine) which are subsequently used as substrates to esterify FC. We found that fluxes through reactions of the TAG-CE pathway were strongly associated with an increased percentage of CE in HDL and decreased TAG levels in LDL and HDL (Table 1, Supplementary Data 3). The pathway was also strongly associated with reduced HDL size, likely driven by a reduction of TAG levels in HDL[52]. While the associations were primarily found in the liver-specific flux map with mediation by liver-expressed enzymes, these pathways are not necessarily constrained to the liver. For example, the hepatic TAG lipase localises to both the liver and blood[53]. Similarly, phospholipids synthesised in the liver may be transferred to HDL in circulation, where they can fuel cholesterol esterification catalysed by the liver-secreted lecithin-cholesterol acyltransferase (LCAT)[52,54].

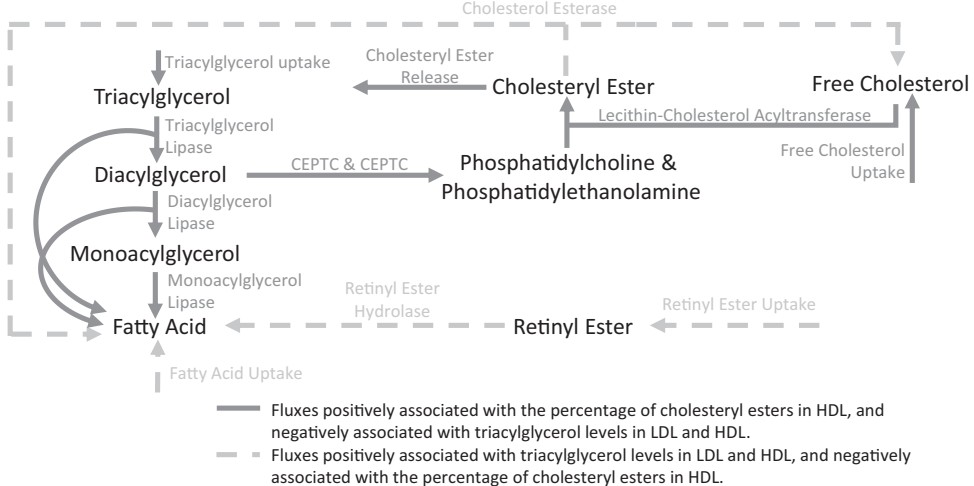

**Fig. 4 | Triacylglycerol to cholesteryl ester pathway in the liver.** Fluxes through reactions of the pathway are negatively associated with triacylglycerol levels in LDL and HDL and positively associated with the percentage of cholesteryl esters in HDL. Fluxes through reactions disrupting the pathway have the opposite effect. Transport processes and some metabolites (e.g., glycerol) have been omitted for clarity. CEPTE ethanolamine phosphotransferase, CEPTC choline phosphotransferase.

Components of the TAG-CE pathway have been the subject of various studies. For example, rare deficiencies in hepatic lipase activity have been linked to increased TAG levels and decreased CE levels in HDL[55]. Similarly, genetic variants in hepatic lipase have been associated with total cholesterol levels in HDL[56,57]. For phospholipids, blocking phosphatidylcholine synthesis has been shown to result in cellular accumulation of TAG both in vitro and in vivo[58,59]. Similarly, LCAT deficiencies have been associated with reduced cholesterol esterification and increased triglycerides in plasma[60,61]. It has also been suggested that cholesterol may be inefficiently esterified by LCAT in patients with CAD, leading to a lower CE to FC ratio[62].

Conversely, flux through reactions disrupting TAG-CE, such as cholesterol esterase, are predicted by our FWAS to have the opposite effect and are associated with increased TAG levels and decreased cholesterol esterification (Table 1, Supplementary Data 3). Among such reactions, there is the hydrolysis of retinyl esters which can act as an alternative source of free fatty acids inhibiting TAG lipase activity (Fig. 4). Retinyl esters are the most abundant form of vitamin A in the human body and are its most common form in diets and vitamin supplements[63]. Dietary retinol is esterified in enterocytes, and most of it is transported to hepatocytes by means of lipoproteins, where it is subsequently hydrolysed and transferred to stellate cells for storage[64]. Notably, the administration of high doses of retinol derivatives has been reported to increase total triglyceride levels and, in some instances, increase total cholesterol in LDL while decreasing total cholesterol in HDL[65-68]. We hypothesise that this occurs because retinyl esters disrupt the hepatic TAG-CE pathway, inhibiting triglyceride lipase and reducing cholesterol esterification, thus reducing the capacity of HDL to collect FC from other lipoproteins such as LDL[52,62,69].

## FWAS identifies metabolic fluxes associated with coronary artery disease

We extended our approach of fluxome-wide analysis to common diseases and performed a multi-tissue FWAS for CAD in UKB. We evaluated the association of the 4300 metabolic fluxes with CAD using Cox regression (Methods), which identified 92 significant associations (FDR-adjusted $P$ value < 0.05 controlling for all tested fluxes). Of such associations, 31 could be replicated with genetically personalised fluxes computed with Recon3D-based models, whereas 61 were specific to the HUMAN1-based models. Liver fluxes yielded the largest number of significant associations with CAD ($N = 32$), followed by fluxes from the adipose tissue ($N = 26$), heart

($N = 15$), brain ($N = 10$), and skeletal muscle ($N = 9$) (Fig. 5; Supplementary Data 4).

The flux of histamine synthesis through histidine decarboxylase was shown to be strongly associated with CAD in adipose tissue with a hazard ratio (HR) per s.d. of log-transformed flux value of 1.060 and a $P$ value of $2.33 \times 10^{-27}$ (Supplementary Data 4). Such association was also detected in the liver, where both the fluxes through histidine uptake (HR = 1.024 per s.d., $P = 1.65 \times 10^{-5}$) and histidine decarboxylation (HR = 1.027 per s.d., $P = 8.60 \times 10^{-7}$) were associated with increased CAD risk. Histamine is an inflammatory mediator synthesised from histidine primarily in mast cells[70], which reside in most tissues, including liver and adipose tissue[71,72]. Histamine has been reported to be associated with atherosclerosis via blood lipids and lipoprotein fractions as well as by promoting inflammation[73-76]. In adipose tissue, polyamine synthesis was also associated with reduced CAD risk (spermidine synthase: HR = 0.9517 per s.d., $P = 6.06 \times 10^{-21}$). Notably, polyamine-rich diets have been established to have a protective effect against cardiovascular disease[77,78]. Moreover, it has recently been determined that polyamines produced by adipose endothelial cells might protect against obesity, a known risk factor for CAD[79], by promoting vascularisation and lipolysis in white adipose tissue[80].

Concerning lipid metabolism, the fluxes through the TAG lipase reactions in adipose tissue (HR = 0.9652 per s.d., $P = 6.48 \times 10^{-11}$), heart (HR = 0.9675 per s.d., $P = 1.06 \times 10^{-9}$) and skeletal muscle (HR = 0.9693 per s.d., $P = 9.83 \times 10^{-9}$) were strongly associated with reduced CAD risk, consistent with the anti-atherogenic effect of lipoprotein lipase activity in these organs[81]. Similarly, the release of free fatty acids from adipose tissue was also associated with reduced CAD risk (e.g., the release of oleic acid: HR = 0.9688 per s.d., $P = 5.09 \times 10^{-19}$ and release of myristic acid: HR = 0.9691 per s.d., $P = 6.64 \times 10^{-9}$). Interestingly, not only is the release of free fatty acids part of normal adipocyte function[21], but it is also a key part of the polyamine-driven signalling cascade in adipose tissue[80]. Conversely, the flux through the phospholipase reaction was associated with increased CAD risk in adipose tissue (HR = 1.027 per s.d., $P = 9.48 \times 10^{-7}$). Notably, phospholipase activities have been suggested to have a causal role in atherosclerosis and have been investigated as potential pharmacological targets to prevent atherosclerosis and CAD[82-85].

The fluxes through several transport processes were also identified as associated with CAD. For instance, histamine transport in the liver appears to be associated with CAD risk in a transport process-specific manner, with histamine transport through uniport being

**Table 1 | Top associations between metabolic fluxes and lipoproteins in the hepatic triacylglycerol to cholesteryl ester pathway**

| Reaction | Blood metabolic feature | Effect size | FDR-adjusted P value (INTERVAL) | FDR-adjusted P value (UKB) |
|---|---|---|---|---|
| Triacylglycerol lipase | HDL_CE_pct | + | $3.48 \times 10^{-31}$ | $2.22 \times 10^{-72}$ |
| | HDL_CE_pct_C | + | $1.86 \times 10^{-73}$ | $<2.225 \times 10^{-308}$ |
| | HDL_size | − | $2.97 \times 10^{-50}$ | $4.67 \times 10^{-292}$ |
| | HDL_TG | − | $9.88 \times 10^{-74}$ | $1.12 \times 10^{-245}$ |
| | LDL_TG | − | $3.37 \times 10^{-78}$ | $3.30 \times 10^{-250}$ |
| Ethanolamine phosphotransferase | HDL_CE_pct | + | $1.49 \times 10^{-29}$ | $5.98 \times 10^{-70}$ |
| | HDL_CE_pct_C | + | $4.39 \times 10^{-74}$ | $<2.225 \times 10^{-308}$ |
| | HDL_size | − | $4.25 \times 10^{-53}$ | $9.52 \times 10^{-307}$ |
| | HDL_TG | − | $2.86 \times 10^{-73}$ | $1.60 \times 10^{-249}$ |
| | LDL_TG | − | $3.99 \times 10^{-79}$ | $1.97 \times 10^{-259}$ |
| Lecithin-cholesterol acyltransferase | HDL_CE_pct | + | $5.45 \times 10^{-22}$ | $1.15 \times 10^{-52}$ |
| | HDL_CE_pct_C | + | $4.60 \times 10^{-46}$ | $1.12 \times 10^{-235}$ |
| | HDL_size | − | $4.24 \times 10^{-33}$ | $2.17 \times 10^{-206}$ |
| | HDL_TG | − | $7.02 \times 10^{-54}$ | $3.70 \times 10^{-171}$ |
| | LDL_TG | − | $9.03 \times 10^{-55}$ | $2.05 \times 10^{-171}$ |
| Cholesteryl ester release | HDL_CE_pct | + | $3.76 \times 10^{-26}$ | $4.17 \times 10^{-62}$ |
| | HDL_CE_pct_C | + | $5.19 \times 10^{-59}$ | $8.75 \times 10^{-291}$ |
| | HDL_size | − | $2.17 \times 10^{-41}$ | $8.02 \times 10^{-254}$ |
| | HDL_TG | − | $1.97 \times 10^{-61}$ | $8.76 \times 10^{-206}$ |
| | LDL_TG | − | $1.23 \times 10^{-66}$ | $1.35 \times 10^{-215}$ |
| Cholesterol esterase | HDL_CE_pct | − | $1.50 \times 10^{-23}$ | $4.31 \times 10^{-57}$ |
| | HDL_CE_pct_C | − | $4.32 \times 10^{-60}$ | $2.10 \times 10^{-264}$ |
| | HDL_size | + | $5.11 \times 10^{-37}$ | $1.29 \times 10^{-226}$ |
| | HDL_TG | + | $3.90 \times 10^{-51}$ | $7.37 \times 10^{-186}$ |
| | LDL_TG | + | $2.92 \times 10^{-60}$ | $2.56 \times 10^{-200}$ |
| Retinyl ester hydrolase | HDL_CE_pct | − | $3.76 \times 10^{-23}$ | $1.04 \times 10^{-53}$ |
| | HDL_CE_pct_C | − | $8.90 \times 10^{-57}$ | $1.24 \times 10^{-261}$ |
| | HDL_size | + | $2.15 \times 10^{-37}$ | $1.87 \times 10^{-227}$ |
| | HDL_TG | + | $3.14 \times 10^{-49}$ | $1.54 \times 10^{-175}$ |
| | LDL_TG | + | $2.19 \times 10^{-57}$ | $3.34 \times 10^{-193}$ |

A complete list of significant associations is provided in Supplementary Data 3. The statistical significance of each flux to lipoprotein associations was evaluated with linear regression (two-tailed *t*-test for flux effect size; Methods). *HDL_CE_pct* cholesteryl esters to total lipids ratio in HDL, *HDL_CE_pct_C* esterified cholesterol to total cholesterol ratio in HDL, *HDL_size* mean diameter for HDL particles, *HDL_TG* triglycerides in HDL, *LDL_TG* triglycerides in LDL.

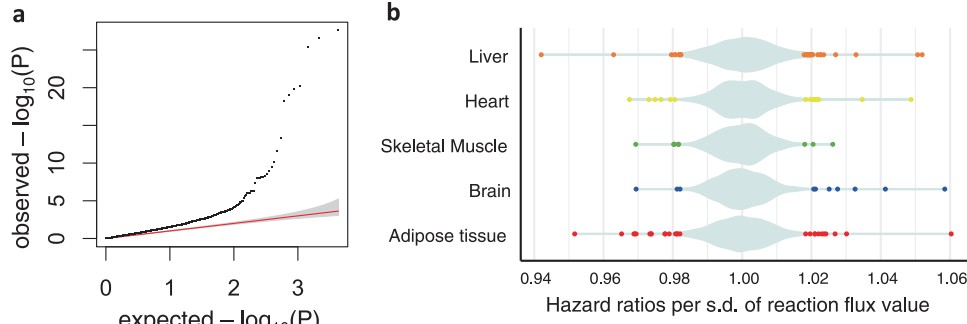

**Fig. 5 | Fluxome-wide association analysis (FWAS) between genetically personalised flux values and coronary artery disease. a** quantile-quantile (QQ) plot of the observed *P* values for associations between flux values and coronary artery disease risk. The red line indicates the expected distribution of *P* values under a uniform distribution (i.e., null hypothesis) and the area coloured in grey shows the 95% confidence intervals of such distribution. **b** Plot of the statistically significant (FDR-adjusted *P* value < 0.05) flux hazard ratios per organ on coronary artery disease risk. The violin plot, coloured in pale azure, shows the distribution of both significant and non-significant hazard ratios. Statistical significance of the association of each flux to coronary artery disease was evaluated with a Cox proportional hazards regression (two-tailed Wald test for flux hazard ratios; Methods).

associated with decreased CAD risk (HR = 0.942 per s.d., $P = 2.60 \times 10^{-28}$) and its antiport with glutathione being associated to increased risk (HR = 1.051 per s.d., $P = 9.77 \times 10^{-20}$). Also in the liver, transport of bilirubin conjugates was associated to decreased CAD risk (transport of bilirubin-monoglucuronoside: HR = 0.9629 per s.d., $P = 2.74 \times 10^{-12}$ and transport of bilirubin-bisglucuronoside: HR = 0.9808 per s.d., $P = 3.49 \times 10^{-4}$). Notably, the transport process of bilirubin-monoglucuronoside is mediated by SLCO1B1, which also mediates the hepatic uptake of statins, enhancing their therapeutic efficacy[86,87]. Interestingly, high levels of total bilirubin in blood have been associated with decreased risk of CAD[88,89], likely mediated by the modulation of arterial diameter and reactivity[90]. In both the brain and heart, the flux of prostaglandin E2 transport was also associated with increased CAD risk (brain: HR = 1.058 per s.d., $P = 4.23 \times 10^{-26}$ and heart: HR = 1.049 per s.d., $P = 5.27 \times 10^{-19}$). Prostaglandin E2 is an inflammatory mediator that promotes inflammation and has been reported to contribute to the development of atherosclerotic lesions[91,92]. Additionally, in the brain, the transport of norepinephrine, a neurotransmitter that can increase blood pressure and may play a role in atherosclerosis[93,94], was also associated with increased risk of CAD (HR = 1.041 per s.d., $P = 4.49 \times 10^{-14}$).

## Discussion

Here, we present a new framework that uses metabolic modelling to leverage the stoichiometric relationships of enzymes in human genome-scale metabolic networks to characterise how genetic variants affect metabolic phenotypes. We achieve this by integrating genetic effects on transcript levels into organ-specific GSMMs and simulating how they propagate and interact into genome-scale flux maps of major human organs. To validate our method, we built organ-specific models for the liver, heart, skeletal muscle, brain, and adipose tissue for over 520,000 individuals from the INTERVAL[35,36] and UKB[37] cohorts, surpassing by more than two orders of magnitude the number of personalised GSMMs built in previous works[30–33]. Association analyses were performed between genetically-personalised flux values and directly measured blood metabolites in both INTERVAL and UKB, identifying many significant and replicable associations. As expected, we found that most blood metabolic features were associated with functionally related flux pathways. Finally, we demonstrate that fluxome-wide analysis can be used to identify putative metabolic drivers of CAD.

With cardiovascular disease being a leading cause of mortality and comorbidity worldwide[95], the identification of specific biochemical reactions linked to CAD using population-scale genomic data is of significant interest to both basic discovery science and the development of therapeutics. Indeed, many of the 92 flux associations we identified involve pathways or metabolites that have been associated with CAD in existing studies, such as histamine[73,76], TAGs[96], or phospholipase activity[82,83,85]. The modulation of some of these fluxes has been explored as therapies for CAD, namely several phospholipase inhibitors[83,84].

Our analysis has several limitations. For instance, as a proof of concept, this study focused on modelling only the five most prominent human organs[40], and thus we can only identify flux to phenotype associations in the liver, heart, skeletal muscle, brain, and adipose tissue. However, given the availability of models to impute tissue- or cell-specific transcript abundance from genotype[13], this analysis can easily be expanded to other tissues and cell types. Indeed, we envision that future applications may select organs for modelling based on the target diseases or phenotypes. Furthermore, the modelling framework presented here is limited to only simulating the effect of genetic variants affecting transcript levels. In the future, it could also be expanded to model the impact of gain or loss of function variants[97] and environmental variables (e.g., diet, lifestyle, and medication) on the personalised flux maps. Additionally, while transcript levels are widely used in genome-scale metabolic modelling[26,29,38], protein levels have a more direct effect on enzymatic activity, and new methods are being

developed to fully integrate them into GSMMs[26,98]. With models to impute the levels of proteins becoming increasingly available[12,15], we expect that the framework for computing genetically personalised fluxes will be extended to integrate the protein layer in the future. Finally, an inherent limitation of our analysis is that it is dependent on the quality of the underlying metabolic networks and their gene-reaction annotations. Indeed, we determined that an important number of the associations between fluxes and blood metabolomics or CAD risk could not be replicated with models based on an earlier reconstruction of human metabolism (i.e., Recon3D[25]). However, with human GSMMs becoming increasingly more well-annotated[26], differences in FWAS results using models built from different genome-scale reconstructions of human metabolism will progressively become more subtle.

Concerning translating genetically personalised models and fluxes to clinical applications, GSMMs have already been established to have utility for drug discovery and repositioning[32,99–102]. Therefore, FWAS may enable identifying fluxes associated with disease states and, by extension, the gene knockdowns or metabolic interventions (e.g., dietary supplements or metabolic inhibitors) to target them. FWAS to blood metabolic features may also help screen for potential adverse side effects of metabolic interventions. For example, we identified that retinyl esters might increase TAG levels and reduce cholesterol esterification in lipoproteins, consistent with reports that administering high doses of vitamin A derivatives results in hypertriglyceridemia and dysregulation of cholesterol levels[65–68]. Furthermore, while it is very early days, personalised fluxes associated with disease risk could also be incorporated into existing risk prediction models, potentially enhancing their predictive capacity.

Overall, this work demonstrates that genome-scale metabolic modelling can contribute to addressing the V2F challenge by characterising how the effects of genetic variants propagate through the metabolic networks of specific human organs.

## Methods

### INTERVAL cohort

INTERVAL is a cohort of approximately 50,000 participants nested within a randomised trial studying the safety of varying the frequency of blood donation (https://clinicaltrials.gov/ct2/show/NCT01610635). Participants were blood donors aged 18 years and older (median 44 years of age; 50% women) recruited between 2012 and 2014 from 25 NHS Blood and Transplant centres[35,36]. Genetically personalised fluxes were computed for the 37,220 individuals with genotype and blood metabolome data that had passed quality control.

Genotyping of INTERVAL samples, their quality control and imputation were performed as previously described:[103] Participants were genotyped in ten batches using Affymetrix UK Biobank arrays. Duplicate samples, samples with extreme heterozygosity or sex mismatch, were removed, and participants of non-European descent were excluded. First- or second-degree relatives (identity-by-descent $\hat{\pi} > 0.187$) were also removed, keeping one sample at random from each pair of close relatives. Genotyped variants were removed if they had a call rate <99%, were monomorphic, or had Hardy-Weinberg equilibrium $P$ value $< 5 \times 10^{-6}$. Variants were subsequently phased using SHAPEIT3, then imputed to the UK10K/1000 Genomes reference panels using the Sanger Imputation Server (https://imputation.sanger.ac.uk).

### UK Biobank

UKB is a cohort of approximately 500,000 participants from the general UK population (https://www.ukbiobank.ac.uk/). Participants were between age 40 and 69 at recruitment (median 58 years of age; 54% women) and accepted an invitation to attend one of the assessment centres that were established across the United Kingdom between 2006 and 2010[37]. Genetically personalised fluxes were computed for the 487,395 individuals in the version 3 release of the UK

Biobank genotype data[104] (https://biobank.ndph.ox.ac.uk/showcase/label.cgi?id=263), which has been imputed to the UK10K/1000 genomes and haplotype reference consortium (HRC)[105] panels.

## Building organ-specific models

For each analysed organ (i.e., adipose tissue, brain, liver, heart, and skeletal muscle), the set of organ-specific metabolic reactions was extracted from the Harvey/Harvetta models (version 1_03c)[30], which contain manually curated metabolic networks for the major organs of the human body. To avoid any gender biases, any reaction present in either the male (Harvey) or female (Harvetta) models was included.

Harvey/Harvetta models were built from the Recon3D human GSMM[25], which has been superseded by HUMAN1[26]. Hence, we performed a liftover to update the Harvey/Harvetta organ-specific models to HUMAN1. Briefly, the IDs of the organ-specific metabolic reactions from the Harvey/Harvetta models were mapped to HUMAN1 (version 1.11.0) using the mapping provided in the HUMAN1 reaction annotations[26]. Subsequently, the resulting set of HUMAN1 reaction IDs was used to assemble organ-specific models from HUMAN1 reactions. Manual curation was used to identify and, when possible, correct gaps and missmaps. Some reactions in the Harvey/Harvetta models that were not present on the base Recon3D and thus could not be mapped to HUMAN1, were also added to the resulting network. These reactions included phospholipase, cholesterol esterase, and extracellular LCAT. Additionally, the side acyl chains of triglycerides and phospholipids were simplified to a stoichiometric mix of 1/3 oleoyl, 1/6 palmitoleoyl, 1/6 palmitoyl, 1/6 stearoyl, 1/6 myristoyl in line with the ratio used in Harvey/Harvetta for non-essential fatty acids[30]. Boundaries for the exchange reactions fluxes (i.e., rates of metabolite uptake or secretion) between each organ-specific model and blood or bile were set as the average bounds of the corresponding reactions in the Harvey and Harvetta models. In some instances, the ranges of metabolite uptake and secretion were further constrained to ensure that they were physiologically relevant. In the brain-specific model, exchange reactions to blood were mapped to the exchange reactions between blood and cerebrospinal fluid defined in Harvey/Harvetta. Such reactions had been defined, taking into consideration the selective permeability of the blood-brain barrier[30]. Thus, only metabolites permeable to this barrier can be exchanged between blood and the brain-specific model. Next, metabolites in blood or bile were made boundary conditions (i.e., assumed constant), allowing each organ subnetwork to function independently. Finally, given that most HUMAN1 reactions lack a name attribute, unnamed reactions in the resulting network were named using their corresponding name in Recon3D. In a number of instances, ambiguously named reactions were manually renamed.

To validate the resulting organ-specific models, we performed flux variability analysis (FVA)[106] to test the capacity of reactions in the networks to carry a significant amount of flux (>10$^{-6}$ mol/day), and 93% were shown to be capable of carrying a significant flux. Furthermore, models were also evaluated against a set of essential metabolic tasks (i.e., tasks all organs are expected to perform to be viable) and organ-specific metabolic tasks obtained from the HUMAN1 repository[26] (Supplementary Data 5). Each organ-specific model was shown to be capable of successfully performing all essential tasks as well as its organ-specific tasks. The resulting organ-specific GSMMs are available on GitHub and permanently archived by Zenodo[107].

Additionally, a set of Recon3D-based organ-specific models were also built. Such models were obtained by applying the steps described above without performing the liftover to HUMAN1.

## Computing organ-specific reference flux maps

The GIM3E algorithm was applied to compute the reference flux map for each organ. The GIM3E algorithm applies a flux minimisation weighted by transcript abundances allowing to find solutions that are enzymatically efficient, consistent with gene expression

data and fulfil a set of metabolic objectives[38] (Supplementary Fig. 1). First, a set of metabolic objectives was defined for each organ representing major metabolic functions that each organ fulfils in the conditions under study (Supplementary Data 6). These were added in each organ subnetwork as lower bounds for flux values through reactions associated with those metabolic objectives. Lower bounds were set relative to the maximum flux feasible through such reactions identified with FVA[106].

Next, organ-specific transcript abundances were obtained as transcripts per million from the GTEx Portal[14] (GTEx Analysis Release V8; dbGaP Accession phs000424.v8.p2; accessed on 05/05/2021) and the average abundance of each transcript in each organ was computed. In the heart, adipose tissue, and brain, there were transcripts abundances measured from multiple source sites. Hierarchical clustering analysis indicated that source sites from each organ were clustered together (Supplementary Fig. 6). Hence, the average of the source sites in each organ was used for the heart, adipose tissue and brain. Average transcript abundances were mapped to the organ-specific subnetworks using the gene reaction annotations of HUMAN1[26]. More in detail, transcript abundances of isoenzymes and enzyme subunits catalysing each reaction or transport process were added and, subsequently, log$_2$ transformed. The resulting values were used as input to apply the flux minimisation weighted by reaction expression[38]:

$$\text{minimise} \sum_i \mathbf{v}_i \cdot \left( \max\left(0, P_{95} - \bar{\mathbf{x}}_{\mathbf{GTEx}_i}\right) + 1 \right) \quad (1)$$

subject to:

$$\mathbf{s} \cdot \mathbf{v} = 0$$

$$\mathbf{lb} \leq \mathbf{v} \leq \mathbf{ub}$$

where, $\mathbf{v}$ is a vector of steady-state flux values; $\bar{\mathbf{x}}_{\mathbf{GTEX}}$ is a vector of average transcript abundances mapped to reactions of the organ-specific network; $P_{95}$ is the 95$^{th}$ percentile of the average transcript abundance values mapped to reactions of the organ-specific network; $\mathbf{s}$ is the stoichiometric matrix. Its product with $\mathbf{v}$ defines the metabolic steady state constraint (i.e., input and output fluxes must be balanced for each metabolite in the network); $\mathbf{lb}$ and $\mathbf{ub}$ are vectors defining the lower and upper bounds of reactions, respectively. The organ-specific metabolic objectives are defined as lower bounds greater than 0 (i.e., constraining such reactions to being active) for the relevant reactions.

Subsequently, FVA was used to identify the feasible flux ranges within 99% of the optimal value of the GIM3E objective function[38]. Finally, the resulting solution space was sampled using the Artificially Centred hit-and-run (ACHR) algorithm[39] implemented into COBRApy[108,109]. ACHR was run with a thinning factor of 1000, and 1000 sets of steady-state flux distributions were computed. The average of those flux samples was used as each organ's reference flux map.

Following the same approach, reference fluxes were also computed for the Recon3D organ-specific models using the gene reaction annotations of Recon3D[25].

## Imputing individual-specific gene expression data

The elastic net models from PredictDB[13] were used to impute organ-specific gene expression levels from individual-level genotypes. These are well-established models that have been extensively validated[13,110–112]. The latest release of PredictDB models, which had been trained with GTEx v8 data, were obtained from https://predictdb.org/. They were used with PLINK2[113] to predict relative transcripts abundances using genotype data from the INTERVAL[35,36] and UKB[37] cohorts. For adipose, brain and heart tissue, the average of the imputed abundances in each source site was used.

## Mapping individual-specific gene expression data to reactions in the model

Imputed individual-specific expression patterns from metabolic genes (i.e., genes coding for enzymes, enzyme subunits, or transmembrane carriers) were mapped to organ-specific models using the gene reaction annotations of HUMAN1[26]. Imputed values were expressed as $\log_2$ fold changes relative to average gene expression in GTEx and mapped to reactions in the organ-specific model considering the relative transcript abundance of isoenzymes and enzyme subunits in GTEx:

$$FC_{R,n} = \frac{\sum_{g \in \mathbf{g_R}} \mathbf{GTEx}_g \cdot 2^{\mathbf{S}_{g,n}}}{\sum_{g \in \mathbf{g_R}} \mathbf{GTEx}_g} \quad (2)$$

where, $\mathbf{S}_{g,n}$ is the organ-specific score for gene $g$ in individual $n$ computed using the elastic net models from PredictDB; $\mathbf{GTEx}_g$ is the average organ-specific gene expression of gene $g$ in GTEx; $\mathbf{g_R}$ are the genes associated with reaction $R$ in the organ-specific network; $\mathbf{FC}_{R,n}$ is the imputed reaction activity fold change for reaction $R$ in individual $n$.

Reaction activity fold changes were also computed for the Recon3D organ-specific models using the gene reaction annotations of Recon3D[25].

## The quadratic metabolic transformation algorithm

Building upon the principle of the metabolic transformation algorithm[101,102], we developed qMTA. qMTA seeks to identify the flux map most consistent with a set of reaction activity fold changes starting from a reference flux distribution (Supplementary Fig. 2). To this end, it minimises the difference between the simulated flux values and the target fluxes (i.e., the product of the flux value in the reference flux distribution and the reaction activity fold change). Additionally, it also minimises the deviation from the reference flux distribution in reactions not mapped to any gene expression fold changes. Furthermore, the two terms of the optimisation function are scaled by the reference flux distribution to prevent biases towards reactions with high flux values.

$$\text{minimise } w \sum_{i \in \mathbf{Ru}} \frac{\left(\mathbf{v}_i^{\mathbf{ref}} - \mathbf{v}_{i,n}^{\mathbf{qMTA}}\right)^2}{\max(|\mathbf{v}_i^{\mathbf{ref}}|, m)} + \sum_{i \in \mathbf{Re}} \frac{\left(\mathbf{v}_i^{\mathbf{ref}} \cdot \mathbf{FC}_{i,n} - \mathbf{v}_{i,n}^{\mathbf{qMTA}}\right)^2}{\max\left(\left(\mathbf{v}_i^{\mathbf{ref}}(\mathbf{FC}_{i,n} - 1)\right)^2, m\right)} \quad (3)$$

Subject to:

$$\mathbf{s}.\,\mathbf{v}_n^{\mathbf{qMTA}} = 0$$

$$\mathbf{lb} < \mathbf{v}_n^{\mathbf{qMTA}} < \mathbf{ub}$$

where, $w$ is the weight given to minimising variation in reactions not mapped to imputed gene expression; $\mathbf{Ru}$ are reactions not mapped to imputed gene expression; $\mathbf{v}^{ref}$ is the flux vector of the reference flux distribution; $\mathbf{v}_{i,n}^{\mathbf{qMTA}}$ is the simulated flux value for reaction $i$ in individual $n$; $m$ is the minimum value allowed for the scaling factor; $\mathbf{Re}$ are reactions mapped to imputed gene expression.

Personalised flux maps computed with qMTA were subsequently $\log_2$ transformed and standardised to zero-mean and unit variance.

Additionally, two hyperparameters in qMTA ($w$ and $m$) were tuned using the regression analysis with blood metabolic features in the INTERVAL cohort. For each simulated organ, a grid search ($w \to [100, 10, 1, 0.1, 0.01]$, $m \to [10^{-6}, 10^{-7}, 10^{-8}, 10^{-9}, 10^{-10}, 10^{-11}, 10^{-12}]$) was performed to identify the parameters that resulted in flux maps with the strongest association with both Nightingale Health and Metabolon HD4 metabolic features. This was measured as the summation of the amount of variance explained ($R^2$) for each blood feature-flux value

pair when testing associations between metabolic fluxes and blood metabolic features. The resulting parameters were subsequently used in the analysis of the samples from UKB. The process was repeated to identify the best set of hyperparameters for the Recon3D-based models.

## Metabolomics

The Nightingale NMR platform quantifies 230 and 249 analytes in INTERVAL and UKB, respectively, including lipoprotein subfractions and ratios, lipids and low molecular weight metabolites (e.g., amino acids)[51]. In INTERVAL, blood samples were profiled with the Nightingale platform at the baseline of the blood donation assay ($N = 37{,}720$). In UKB, metabolite concentrations were determined in 117,981 participants at baseline assessment and 5141 participants at repeat assessment, among which there were 1427 participants with measurements at both time points. For participants with measurements at both baseline and repeat assessment, the measurement at baseline assessment was used[114]. Values were adjusted for technical covariates using the ukbnmr R package[114] and subsequently regressed for age, sex, BMI, and the first 5 PCs of genetic ancestry. Composite biomarkers and ratios were recomputed after adjustment, including 98 and 76 additional biomarker ratios in INTERVAL and UKB, respectively, not provided by the Nightingale platform. Metabolic features not present in both INTERVAL and UKB were excluded from downstream analyses. Likewise, 68 features with markedly distinct variance between INTERVAL and UKB ($|\log2(sd_{INTERVAL}/sd_{UKB})| > \log2(2.5)$) were also excluded. Finally, acetate was excluded due to a large number of NA (>75%) in INTERVAL. Subsequently, measures were standardised to zero-mean and unit variance.

The Metabolon HD4 assay measures ~1000 metabolites (~700 named, ~300 unknown), including lipids, xenobiotics, amino acids and energy-related metabolites. A subset of INTERVAL participants (N=8,115) had their blood profiled with this assay, predominantly using baseline blood samples. Nineteen features were excluded due to a large number of NA (>75%). Values were regressed against technical covariates age, sex, BMI, and the first 5 PCs of genetic ancestry. Subsequently, measures were standardised to zero mean and unit-variance.

## Testing associations between metabolic fluxes and blood metabolic features

Due to the linear nature of many metabolic pathways, some flux values were highly intercorrelated (Fig. S3). To remove reaction flux pairs with a strong correlation, for each pair of reaction flux values with $\rho > 0.9$, the feature with the largest mean absolute correlation to other flux values was removed[115]. Likewise, both the Nightingale and Metabolon platforms had some metabolic features with strong correlations, and those features with $\rho > 0.75$ were removed using the same approach used for reaction fluxes. Overall, 4300 scaled flux values and 57 Nightingale Health and 718 Metabolon HD4 blood metabolic features were selected to perform FWAS.

Then, the association of each metabolic feature to each personalised flux value was evaluated using linear regression (Supplementary Fig. 2).

$$\text{Met} = a_{\text{Met},i} \cdot \mathbf{v}_i^{\mathbf{qMTA}} + \varepsilon \quad (4)$$

where, Met are the measured levels of a blood metabolic feature; $a_{\text{Met},i}$ is the effect size of flux $i$ on Met; $\varepsilon$ is the residual.

Statistical significance was evaluated with a $t$-test (two-tailed) on effect sizes. The resulting $P$ values were adjusted for multiple testing against all evaluated blood metabolic features–reaction flux pairs using the Benjamini and Yosef Hochberg (i.e., FDR) method.

To evaluate the association between metabolic fluxes computed with Recon3D-based models and blood metabolic features, the set of

4300 uncorrelated flux values in HUMAN1 was mapped to equivalent reactions in the Recon3D-based models. This set of flux values was then used to perform FWAS to the same set of 57 Nightingale Health and 718 Metabolon HD4 blood metabolic features as the HUMAN1 analysis.

## Classes of blood metabolic features

Nightingale/Metabolon platforms provide sets of Groups/Sub-pathways to stratify metabolic features. We harmonised both annotations systems to define a set of curated groups that could be applied to both Nightingale and Metabolon features. For instance, the Metabolon features annotated to "Glycerolipid Metabolism" and "Phospholipid metabolism", and the Nightingale features annotated to "Phospholipids" were all assigned to the curated group "Glycerides and phospholipids". Some Metabolon features were not annotated (i.e., unknown) and could not be assigned to any curated group. Unknown features were included in the FWAS but omitted from the enrichment analysis. Fisher's exact test (one-sided) was used to identify metabolite classes enriched in features with significant association to personalised flux values relative to the set of all uncorrelated blood metabolic features. An FDR-adjusted significance threshold of $P < 0.05$ was applied to control for all tested classes of blood metabolic features across all organs.

## Reaction systems

Subsystem annotations for reactions were obtained from the HUMAN1 model[26]. As some subsystems contained a low number of reactions, functionally related subsystems were merged into larger reaction systems. For instance, the purine metabolism, pyrimidine metabolism and nucleotide metabolism subsystems were aggregated into a reaction system termed nucleotide metabolism. Additionally, transport processes (i.e., annotated in the transport or exchange reactions subsystems) were assigned a system based on the specific metabolites being transported in each process. Briefly, we first assigned a system to each metabolite based on the most frequent reaction system annotation in the reactions in which it participates. For instance, alanine was assigned to the system "amino acid metabolism" since it was the system annotated most in reactions in which alanine participated. Next, each transport process/exchange reaction in HUMAN1 was assigned the system most numerous in the metabolites being transported. For the purpose of this assignment, metabolites that are often cofactors in transport processes (e.g., $Na^+$, $K^+$, $H^+$, and ATP/ADP) were set to give less weight than other metabolites. For instance, the alanine-sodium symporter (alanine[e] + $Na^+$[e] → alanine[c] + $Na^+$[c]) was assigned to the system "amino acid metabolism" as alanine (system: amino acid metabolism) was given more weight than $Na^+$ (system: Miscellaneous). Reaction systems are solely used as annotations and have no influence on network stoichiometry or genetically personalised flux values.

Fisher's exact test (one-sided) was used to identify reaction systems enriched in reactions with significant association with blood metabolic features relative to the set of all evaluated reactions in each organ. An FDR-adjusted significance threshold of $P < 0.05$ was applied to control for all tested systems across all organs.

## Testing associations between metabolic fluxes and coronary artery disease

Using PheWAS Catalogue (version 1.2), we used the WHO International Classification of Diseases (ICD) diagnosis codes in versions 9 (ICD-9) and 10 (ICD-10) of Phecode 411.4 for CAD case definition in UKB. In detail, we searched for the presence of any of the constituent ICD-9/10 codes in linked health records (including in-patient Hospital Episode Statistics data, and primary and secondary cause of death information from the death registry) and converted the earliest coded date to the age of phenotype onset. Individuals without any codes for CAD were recorded as controls and censored according to the maximum follow-up of the health linkage data (January 31, 2020) or the date of death.

We recorded 34,121 events of CAD and 428,669 controls in UKB, which were used to evaluate the association of genetically personalised fluxes to CAD risk. Association was tested using an age-as-time-scale Cox proportional hazards regression. The Cox models were stratified by sex and adjusted by genotyping array, 10 genetic PCs, BMI and smoking status and fitted using the CoxPHFitter function from the lifelines package for python[116]. The significance of the flux to CAD risk associations was evaluated with a two-tailed Wald test for the flux HRs. The resulting $P$ values were adjusted for multiple testing against all tested fluxes using the Benjamini and Yosef Hochberg (i.e., FDR) method.

## Reporting summary

Further information on research design is available in the Nature Portfolio Reporting Summary linked to this article.

## Data availability

The data from the INTERVAL[35,36] and UK Biobank[37] cohorts is under restricted access as it contains potentially identifying and sensitive patient information. It can be accessed by making a reasoned request to the INTERVAL coordination centre (https://www.intervalstudy.org.uk) and UKB (https://www.ukbiobank.ac.uk/), respectively. Response times from the data access committees are typically under one month. The summary statistics for the FWAS to blood metabolic features and CAD are provided in Supplementary Data 2, Supplementary Data 3 and Supplementary Data 4. The organ-specific genome-scale metabolic models generated in this work are available on the cobrafunctions GitHub repository, which is permanently archived by Zenodo[107]. HUMAN1[26] (version 1.11.0) can be obtained from the Human-GEM GitHub repository. The Harvey and Harvetta models (1_03c) are available in the Supporting Information of reference 30. The elastic net PredictDB models (GTEx v8) models[13] are available at https://predictdb.org. The GTEx[14] gene expression data (GTEx Analysis Release V8; dbGaP Accession phs000424.v8.p2) can be obtained from https://gtexportal.org.

## Code availability

The code used to generate personalised organ-specific flux maps from imputed gene expression data is available on GitHub and permanently archived by Zenodo[107]. qMTA requires the proprietary solver CPLEX(2.6 or newer), which is freely available to academic users.

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

## Acknowledgements

Participants in the INTERVAL trial were recruited with the active collaboration of NHS Blood and Transplant (www.nhsbt.nhs.uk), which has supported fieldwork and other elements of the trial. DNA extraction and genotyping were co-funded by the National Institute for Health and Care Research (NIHR), the NIHR BioResource (http://bioresource.nihr.ac.uk) and the NIHR Cambridge Biomedical Research Centre (BRC) (no. BRC-1215-20014)*. Nightingale Health NMR assays were funded by the European Commission Framework Programme 7 (HEALTH-F2-2012-279233). Metabolon Metabolomics assays were funded by the NIHR BioResource and the NIHR Cambridge Biomedical Research Centre (BRC-1215-20014)*. The academic coordinating centre for INTERVAL was supported by core funding from the NIHR Blood and Transplant Research Unit in Donor Health and Genomics (no. NIHR BTRU-2014-10024), NIHR BTRU in Donor Health and Behaviour (NIHR203337), UK Medical Research Council (MRC) (no. MR/L003120/1), British Heart Foundation (nos SP/09/002, RG/13/13/30194 and RG/18/13/33946) and the NIHR Cambridge BRC (no. BRC-1215-20014)*. *The views expressed are those of the author(s) and not necessarily those of the NIHR, NHSBT or the Department of Health and Social Care. A complete list of the investigators and contributors to the INTERVAL trial is provided in ref. 36. The academic coordinating centre would like to thank blood donor centre staff and blood donors for participating in the INTERVAL trial. This work was supported by Health Data Research UK, which is funded by the UK MRC, Engineering and Physical Sciences Research Council (EPSRC), Economic and Social Research Council, Department of Health and Social Care (England), Chief Scientist Office of the Scottish Government Health and Social Care Directorates, Health and Social Care Research and Development Division (Welsh Government), Public Health Agency (Northern Ireland), British Heart Foundation and Wellcome. This research has been conducted using the UK Biobank Resource under Application 7439. This work was performed using resources provided by the Cambridge Service for Data-Driven Discovery (CSD3) operated by the University of Cambridge Research Computing Service (www.csd3.cam.ac.uk), provided by Dell EMC and Intel using Tier-2 funding from the Engineering and Physical Sciences Research Council (capital grant EP/P020259/1), and DiRAC funding from the Science and Technology Facilities Council (www.dirac.ac.uk). C.F. is funded Health Data Research UK. S.R. is funded by the NIHR Cambridge Biomedical Research Centre (BRC-1215-20014). S.L. is supported by a Canadian Institutes of Health Research postdoctoral fellowship (MFE-171279). E.P. was funded by the EU/EFPIA Innovative Medicines Initiative Joint Undertaking BigData@Heart grant 116074 and the NIHR BTRU in Donor Health and Genomics (NIHR BTRU-2014-10024) and is funded by the NIHR BTRU in Donor Health and Behaviour (NIHR203337). E.E.D. is supported by the Wellcome Trust grant (206194, 108413/A/15/D). J.D. holds a British Heart Foundation Professorship and an NIHR Senior Investigator Award. M.I. is supported by the Munz Chair of Cardiovascular Prediction and Prevention and the NIHR Cambridge Biomedical Research Centre (BRC-1215-20014). M.I. was also supported by the UK Economic and Social Research 878 Council (ES/T013192/1).

## Author contributions

Conceptualisation: C.F., C.Y., and M.I.; Formal Analysis: C.F., Y.X., S.C.R., and M.I.; Investigation: S.C.R., S.A.L., E.P., A.P.N., E.E.D., D.J.R., D.S.P., E.D.A., J.D., A.S.B., and M.I.; Writing—original draft: C.F., C.Y., and M.I.; Supervision: C.Y. and M.I.; All authors reviewed and approved the final paper.

## Competing interests

A.S.B. has received grants (outside of this work) from AstraZeneca, Bayer, Biogen, BioMarin, Bioverativ, Merck, Novartis, Regeneron, and Sanofi. J.D. serves on scientific advisory boards for AstraZeneca, Novartis, and UK Biobank, and has received multiple grants from academic, charitable and industry sources outside of the submitted work. During the preparation of the paper, D.S.P. became a full-time employee of AstraZeneca. The remaining authors declare no competing interests.
