## [Peer Review File · Nature Communications]

Genetically personalised organ-specific metabolic models in health and diseaseREVIEWER COMMENTS

Reviewer #1 (Remarks to the Author):

Fouget et al. presented an interesting study on genetically personalized metabolic models and performed a fluxome-wide association study (FWAS) to test the association between organ-specific flux values and directly measured blood metabolite levels.

The comprised of data from INTERVAL and UK Biobank cohorts.

The following limitations need to be addressed in the manuscript:

- The authors do not discuss the medications prescribed to individuals in this study and also how medication might affect the flux state. There are 50,000 participants in INTERVAL and 500,000 participants in UK Biobank, but the authors report that personalised organ-specific flux maps were built for 37,220 and 487,395 individuals from the INTERVAL and UKB cohorts. Why is this number different than the total number of individuals in the cohort? Was there a filtration step not specified in the methods?
- It is unclear whether individuals in this study donated blood more than once? It will be helpful if the authors can specify frequency of blood donation by participants.
- The authors considered metabolites in blood or bile ducts for setting boundary conditions. How about the boundary condition for brain? The metabolites that can pass through the blood brain barrier are very selective and need to be considered for boundary conditions for brain.
- The authors have defined a set of metabolic requirements representing metabolic functions that each organ fulfils in physiological conditions. But, in addition to these metabolic requirements, the authors also need to test the models for the metabolic tasks that they pass. These tasks have been described in the Recon3D paper (PMID: 29457794).
- I am not quite satisfied with the rationale behind reassigning reactions from transport subsystem to the reaction subsystem most frequent in metabolite transport (lines 490-491). Won't this result in a proportionately higher number of reactions in a subsystem where the number of metabolites is higher? Also, how did the authors account for the transport reaction for common metabolites in the system? Is that the reaction, why the principal component analysis for organs reflected transport reactions with higher loadings in liver, skeletal muscle, and brain (lines 132-144)?
- The authors do not clearly state total number of individuals with coronary artery disease in the cohort? Distribution of individuals in both the cohorts with respect to the major categories in ICD will be helpful to ascertain the diseases to focus on.
- It is not clear how the authors are using the genotyping data from INTERVAL and UK Biobank cohorts? It seems that GTEx data is being predominantly used as an input.
- Is the code for qMTA provided in a public repository?
- The authors should discuss limitations of their study and also how their work can be used for translational study.

Reviewer #2 (Remarks to the Author):

This paper describes a large effort on generating personalized organ-specific metabolic model. Using a large cohort of subjects with their genome sequenced the authors predict the transcripts and then use this to generate organ specific genome-scale metabolic models (GEMs). I very much like this work and I think it represents a major step forward, and in particular it is interesting that the authors use calculated fluxes to identify new markers (what they call FWAS). However, the authors needs to validate their findings. I have two major concerns:

1) The authors use PredictDB to impute organ-specific transcript abundances and use these for model generation. PredictDB is from a separate publication (in 2015) and I do not know how the data generated are validated, but the authors should at least validate their findings here by e.g. using data from biopsies where there is both genome sequence data and transcript data.

2) The authors say they are using the Recon family of GEMs. What does this mean? Which of the models? Why not use the far superior models presented more recently, i.e. Human1, which has been

shown to have much less gaps and errors, and be much better in predicting fluxes. The latter becomes in particular important as the authors aim to calculate fluxes. This also leads to another criticism, namely why not use the GECKO approach for modeling. This has been shown again to significantly improve flux predictions. In fact I think the resolution of the authors findings could be significantly improved by doing this. I am aware that this is basically asking the authors to redo a lot of their analysis, but it is not good for the research community that old models that have been shown to be poor for flux simulations are used in new and exciting studies like the one presented here.

Reviewer #1 (Remarks to the Author):

Foguet et al. presented an interesting study on genetically personalized metabolic models and performed a fluxome-wide association study (FWAS) to test the association between organ-specific flux values and directly measured blood metabolite levels.

The comprised of data from INTERVAL and UK Biobank cohorts.

The following limitations need to be addressed in the manuscript:

- **The authors do not discuss the medications prescribed to individuals in this study and also how medication might affect the flux state.**

We thank the reviewer for raising this and their overall positive assessment of our study. In this work, personalized fluxes are computed from genotype data alone, so they do not reflect the specific medication prescribed to each individual or other environmental factors. This is intentional as we aim to analyse how genetic variants cause propensity towards a given flux state and its implications in health and disease. However, we agree that medication usage could be incorporated into flux estimates reflecting both the genetic and environmental state of the individual; therefore, we have added this as part of the Discussion:

Our analysis has several limitations.(...) the modelling framework presented here is limited to only simulating the effect of genetic variants affecting transcript levels. In the future, it could also be expanded to model the impact of gain or loss of function variants(Jamshidi and Palsson 2006) and environmental variables (e.g., diet, lifestyle, medication) on the personalized flux maps.

- **There are 50,000 participants in INTERVAL and 500,000 participants in UK Biobank, but the authors report that personalized organ-specific flux maps were built for 37,220 and 487,395 individuals from the INTERVAL and UKB cohorts. Why is this number different than the total number of individuals in the cohort? Was there a filtration step not specified in the methods?**

We are grateful for the feedback of the reviewer so that we may clarify the filtration steps. In UKB, we computed genetically personalized fluxes for the 487,395 individuals with genotype data that had passed quality control (see Methods and ref 102 in the revised manuscript). In INTERVAL, the focus was on the association of fluxes to blood metabolome, so we computed genetically personalized fluxes for the 37,220 individuals with both genotype and blood metabolome measures. We have now clarified these steps in the Methods subsections describing the INTERVAL and UK Biobank cohorts.

It is worth noting that fluxes are computed independently for each individual, so excluding INTERVAL participants without metabolomics measures has no effect on fluxes simulated on the remaining INTERVAL participants.

- **It is unclear whether individuals in this study donated blood more than once? It will be helpful if the authors can specify frequency of blood donation by participants.**

Participants in both cohorts did donate blood more than once. Indeed, the goal of INTERVAL was to determine the optimal interval between blood donations(Moore et al. 2014; Di Angelantonio et al. 2017). Likewise, in UKB, a subset of 20,000 participants was selected for a repeat assessment(Sudlow et al. 2015). However, the frequency of blood donation is not a factor in the FWAS to blood metabolomics as the vast majority of metabolomics measures were taken at baseline, and a single set of measures was used for each individual.

We have clarified this in the Methods section by adding the following:

The Nightingale NMR platform quantifies 230 and 249 analytes in INTERVAL and UKB, respectively, including lipoprotein subfractions and ratios, lipids and low molecular weight metabolites (e.g., amino acids)(Julkunen et al. 2021). In INTERVAL, blood samples were profiled with the Nightingale platform at the baseline of the blood donation assay (N= 37,720). In UKB, metabolite concentrations were determined in 117,981 participants at baseline assessment and 5,141 participants at repeat assessment, among which there were 1,427 participants with measurements at both time points. For participants with measurements at both baseline and repeat assessment, the measurement at baseline assessment was used(Ritchie et al. 2021). (...)

The Metabolon HD4 assay measures ~1000 metabolites (~700 named, ~300 unknown), including lipids, xenobiotics, amino acids and energy-related metabolites. A subset of INTERVAL participants (N=8,115) had their blood profiled with this assay, predominantly using baseline blood samples.

- **The authors considered metabolites in blood or bile ducts for setting boundary conditions. How about the boundary condition for brain? The metabolites that can pass through the blood brain barrier are very selective and need to be considered for boundary conditions for brain.**

We appreciate the concern of the reviewer. Boundary metabolites and exchange reactions for each organ-specific model were extracted from the Harvey/Harvetta multiorgan models(Thiele et al. 2020). For the brain, Thiele *et al.* had done extensive research into which metabolites could be transported across the blood-brain barrier (see Table EV9 in their article), and only such metabolites were set as boundary conditions for the brain(Thiele et al. 2020). Thus, the selective permeability of the blood-brain barrier is accounted for in our brain-specific model. We have added the following to the Methods to clarify:

In the brain-specific model, exchange reactions to blood were mapped to the exchange reactions between blood and cerebrospinal fluid defined in Harvey/Harvetta. Such reactions had been defined taking into consideration the selective permeability of the blood-brain barrier(Thiele et al. 2020). Thus, only metabolites permeable to this barrier can be exchanged between blood and the brain-specific model. Next, metabolites in blood or bile were made boundary conditions (i.e., assumed constant), allowing each organ subnetwork to function independently.

- **The authors have defined a set of metabolic requirements representing metabolic functions that each organ fulfils in physiological conditions. But, in addition to these metabolic requirements, the authors also need to test the models for the metabolic tasks that they pass. These tasks have been described in the Recon3D paper (PMID: 29457794).**

We have now added a validation step where we test the capacity of the organ-specific models to fulfill essential tasks (i.e., tasks that all organs are expected to be able to perform to be viable) and organ-specific tasks. In response to the input of the second reviewer, we have switched from using Recon3D-based models to HUMAN1-based models, so we have obtained the tasks to evaluate the models from the Human1 repository(Robinson et al. 2020). We have the following sentences in the Methods section outlining the validation:

Furthermore, models were also evaluated against a set of essential metabolic tasks (i.e., tasks all organs are expected to perform to be viable) and organ-specific metabolic tasks obtained from the HUMAN1 repository(Robinson et al. 2020) (Table S5). Each organ-specific model was shown to be capable of successfully performing all essential tasks as well as its organ-specific tasks.

- I am not quite satisfied with the rationale behind reassigning reactions from transport subsystem to the reaction subsystem most frequent in metabolite transport (lines 490-491). Won't this result in a proportionately higher number of reactions in a subsystem where the number of metabolites is higher? Also, how did the authors account for the transport reaction for common metabolites in the system? Is that the reaction, why the principal component analysis for organs reflected transport reactions with higher loadings in liver, skeletal muscle, and brain (lines 132-144)?

We thank the reviewer for querying this. We use reaction subsystems/systems annotations to annotate reactions in FWAS results, particularly for Figure 3 and its associated discussion. It is worth noting that reaction subsystems/systems are solely used as annotations and have no influence on network stoichiometry or genetically personalised flux values. To address Reviewer 2's comments, we have done a significant rework on reaction annotation as part of the switch to HUMAN1 models. First, we are now using the HUMAN1 subsystem annotations rather than the Recon3D annotations. In the original manuscript, similar subsystems were aggregated into large subsystems (as was stated in the Methods). To prevent confusion, in the revised manuscript, we now term these "large subsystems" as reaction systems (e.g., the subsystems pyrimidine synthesis and purine synthesis are assigned to the reaction system nucleotide metabolism). To summarize the results in Figure 3, we use reaction systems, but the supplementary tables (Table S1, Table S2, Table S3, Table S4) also include the subsystem annotation. The process is detailed in the Methods of the revised manuscript.

Regarding transport processes or exchange reactions, they are assigned to the most relevant system based on the metabolites being transported by each transport process. We apologize for the confusion, as we now realize it was worded vaguely in the original manuscript. The explanation of how this is achieved has been significantly expanded in the Methods:

Additionally, transport processes (i.e., annotated in the transport or exchange reactions subsystems) were assigned a system based on the specific metabolites being transported in each process. Briefly, we first assigned a system to each metabolite based on the most frequent reaction system annotation in the reactions where it participates. For instance, alanine was assigned to the system "amino acid metabolism" since it was the system annotated most in reactions where alanine participated. Next, each transport process/exchange reaction in HUMAN1 was assigned the system most numerous in the metabolites being transported. For the purpose of this assignment, metabolites that are often cofactors in transport processes (e.g., Na⁺, K⁺, H⁺, ATP/ADP) were set to give less weight than other metabolites. For instance, the alanine-sodium symporter (alanine[e] + Na⁺[e] → alanine[c] + Na⁺[c]) was assigned to the system "amino acid metabolism" as alanine (system: amino acid metabolism) was given more weight than Na⁺ (system: Miscellaneous). Reaction systems are solely used as annotations and have no influence on network stoichiometry or genetically personalised flux values.

This approach will indeed result in a higher number of reactions in systems involving a large number of metabolites with transport processes. For the purposes of Figure 3 and the related discussion, this is taken into account by using the Fisher test to identify systems enriched in significant associations relative to system size and all tested reactions (as stated in Methods).

The rationale for reassigning transport and exchange reactions is to visualize and interpret the functional relationship between fluxes and blood metabolites in the FWAS to blood metabolome (Figure 3). Our concern was that the subsystem transport process and exchange reactions did not adequately convey any clear relationship to blood metabolite classes as they would contain transport processes involved in the transport of all types of metabolites. For instance, knowing that a reaction in the subsystem "transport process" is associated with bile acid levels in blood is not as informative as knowing that such a reaction involves the transport of a bile acid. Hence, we assigned transport

processes to the reaction system most relevant for the metabolite being transported (e.g., bile acid metabolism in the above-mentioned example).

Regarding PCA loadings, while the discussion uses reaction systems when appropriate, it is not focused solely on reaction systems/subsystems and instead aims to provide an overview of the top reactions in each organ with more granularity than those afforded by the subsystem/system annotations[†]. Regarding the presence of amino acid transport processes in the top loadings on skeletal muscle and heart, we believe that it reflects both the importance of protein metabolism and AA transport in their metabolic networks (Drake et al. 2012; Dickinson and Rasmussen 2013) as well as that those processes are subject to genetically driven-variation in those tissues.

- **The authors do not clearly state total number of individuals with coronary artery disease in the cohort? Distribution of individuals in both the cohorts with respect to the major categories in ICD will be helpful to ascertain the diseases to focus on.**

We have now included the number of CAD cases and controls in UKB. The following sentence has been added in Methods:

We recorded 34121 events of CAD and 428669 controls in UKB, which were used to evaluate the association of genetically personalised fluxes to CAD risk.

We did not test the association to CAD risk in the INTERVAL cohort due to the low number of CAD cases in this cohort (due to the stringently healthy inclusion criteria necessary for blood donors).

Additionally, in this work, we chose to focus on CAD as proof of concept, but we plan to expand the analysis to other diseases or conditions in the future. We agree with the reviewer that the distribution of individuals with specific ICD codes will be the major factor in deciding which diseases to analyse. We thank the reviewer for this suggestion.

- **It is not clear how the authors are using the genotyping data from INTERVAL and UK Biobank cohorts? It seems that GTEx data is being predominantly used as an input.**

We apologise if the methodology wasn't sufficiently clear. Below, we briefly summarise for the reviewer. Genotype data is used to impute personalised transcript levels in the analysed individuals, which are then used to compute personalised flux values. Given that imputed organ-specific transcript levels are relative to the average expression of each transcript in the general population, GTEx data is used to provide a baseline for both transcripts and fluxes.

More in detail, genotyping data from INTERVAL and UK Biobank is used to impute personalized transcript levels using PredictDB models (Gamazon et al. 2015). Briefly, such models aggregate the effect of genetic variants on gene expression and allow to estimate gene expression values of a set of genetically controlled genes from the variants present in each individual's genome. Incidentally, such models had been trained using GTEx data by Gamazon et al. We have reworded the description of the PredictDB models to reflect this:

[†] For instance, we state that in skeletal muscle and heart some top PCs were associated with reactions related to fatty acid β -oxidation, because inspection of the reactions reveals that they are linked to it. Had we used systems instead, we would have to state that the reactions belong to fatty acid metabolism, which is less specific. Conversely, had we used subsystems, we would have to state that the reactions belong to "Beta oxidation of even-chain fatty acids (mitochondrial)", "Carnitine shuttle (cytosolic)", "Carnitine shuttle (mitochondrial)", "Exchange/demand reactions" and "Fatty acid oxidation" which can be both too specific and a bit redundant.

The elastic net models from PredictDB(Gamazon et al. 2015) were used to impute organ-specific gene expression levels from individual-level genotypes. These are well-established models that have been extensively validated(Gamazon et al. 2015; Li et al. 2018; Tavares et al. 2021; Hale et al. 2021). The latest release of PredictDB models, which had been trained with GTEx v8 data, were obtained from <https://predictdb.org/>. They were used with PLINK2(Chang et al. 2015) to predict relative transcripts abundances using genotype data from the INTERVAL(Moore et al. 2014; Di Angelantonio et al. 2017) and UKB(Sudlow et al. 2015) cohorts.

The resulting imputed transcript data are mapped to reactions in the organ-specific subnetworks as putative reaction activity fold changes relative to the average organ-specific transcript expression in GTEx(GTEx Consortium 2013). Briefly, GTEx data is here used to account for the relative expression of all isoenzymes and enzyme subunits participating in a given reaction. This allows giving each genetically imputed variation of transcript levels a weight proportional to the average relative contribution of its associated isoenzyme/subunit to a given reaction.

In parallel, we also use average transcript data from GTEx as input to compute the reference flux distribution in each organ (which we assume to represent an average flux distribution in the general population). We integrate this average flux distribution with the personalized transcript variation imputed from genotype data to compute personalized fluxes.

The process is summarized in Figure 1, which we have included below.

Fig. 1: Framework for computing organ-specific personalised genome-scale flux maps from genotype data and performing fluxome-wide association study (FWAS). First, we extract the organ-subnetworks from the Harvey/Harvetta multiorgan models, which were built from RECON3D,

and we perform a liftover to update them to HUMAN1, the most recent human GSMM. Then, a reference flux map is computed for each organ using the GIM3E algorithm to integrate average transcript abundances and organ-specific metabolic functions into the organ-specific metabolic subnetwork. In parallel, personalised organ-specific transcript abundances are imputed from genotype data of the INTERVAL and UKB cohorts using the models from PredictDB. Next, the quadratic metabolic transformation algorithm (qMTA) is used to integrate the organ-specific transcript abundances and reference flux distribution and compute personalised organ-specific metabolic flux maps. The resulting flux maps can be used to perform FWAS to complex traits or diseases such as blood metabolic features or coronary artery disease. A hypothetical representation of an organ-specific solution space, reference flux distribution, and personalised flux distributions is shown for a reaction network with three fluxes (v_x , v_y and v_z).

- **Is the code for qMTA provided in a public repository?**

Yes, as stated in the Data and Code Availability section of the manuscript, the code for qMTA is available on GitHub (<https://github.com/cfoquet/cobrafunctions>).

- **The authors should discuss limitations of their study and also how their work can be used for translational study.**

We have followed the reviewer's advice and added the following paragraph discussing the limitations of our work:

Our analysis has several limitations. For instance, as a proof of concept, this study focused on modelling only the five most prominent human organs (Gallagher, Chung, and Akram 2013), and thus we can only identify flux to phenotype associations in the liver, heart, skeletal muscle, brain, and adipose tissue. However, given the availability of models to impute tissue- or cell-specific transcript abundance from genotype (Gamazon et al. 2015), this analysis can easily be expanded to other tissues and cell types. Indeed, we envision that future applications may select organs for modelling based on the target diseases or phenotypes. Furthermore, the modelling framework presented here is limited to only simulating the effect of genetic variants affecting transcript levels. In the future, it could also be expanded to model the impact of gain or loss of function variants (Jamshidi and Palsson 2006) and environmental variables (e.g., diet, lifestyle, medication) on the personalized flux maps. Additionally, while transcript levels are widely used in genome-scale metabolic modelling (Schmidt et al. 2013; Jamialahmadi et al. 2019; Robinson et al. 2020), protein levels have a more direct effect on enzymatic activity, and new methods are being developed to fully integrate them into GSMMs (Robinson et al. 2020; Sánchez et al. 2017). With models to impute the levels of proteins becoming increasingly available (Xu et al. 2022; Wingo et al. 2021), we expect that the framework for computing genetically personalized fluxes will be extended to integrate the protein layer in the future. Finally, an inherent limitation of our analysis is that it is dependent on the quality of the underlying metabolic networks and their gene-reaction annotations. Indeed, we determined that an important number of the associations between fluxes and blood metabolomics or CAD risk could not be replicated with models based on an earlier reconstruction of human metabolism (i.e., Recon3D (Brunk et al. 2018)). With human GSMMs becoming increasingly more well-annotated (Robinson et al. 2020), differences in FWAS results using models built from different genome-scale reconstructions of human metabolism will progressively become more subtle.

We believe that the second to last paragraph in the discussion was already centred on how personalized fluxes and FWAS might be translated to a clinical setting. Still, we have added some additional sentences to make it more explicit:

Concerning translating genetically personalized models and fluxes to clinical applications, GSMMs have already been established to have utility for drug discovery and repositioning (Agren et al. 2014; Folger et al. 2014; Raškevičius et al. 2018; Yizhak et al. 2013). Therefore, FWAS, may enable identifying fluxes associated with disease states and, by extension, the gene knockdowns or metabolic interventions (e.g., dietary supplements or metabolic inhibitors) to target them. FWAS to blood metabolic features may also help screen for potentially adverse side effects of metabolic interventions. For example, we identified that retinyl esters might increase TAG levels and reduce cholesterol esterification in lipoproteins, consistent with reports that administering high doses of Vitamin A derivatives results in hypertriglyceridemia and dysregulation of cholesterol levels (Murray, Gilgor, and Lazarus 1983; Vahlquist et al. 1985; Bershady et al. 1985; Redlich et al. 1999). Furthermore, while it is very early days, personalized fluxes associated with disease risk could also be incorporated into existing risk prediction models, potentially enhancing their predictive capacity.

Reviewer #2 (Remarks to the Author):

This paper describes a large effort on generating personalized organ-specific metabolic model. Using a large cohort of subjects with their genome sequenced the authors predict the transcripts and then use this to generate organ specific genome-scale metabolic models (GEMs). I very much like this work and I think it represents a major step forward, and in particular it is interesting that the authors use calculated fluxes to identify new markers (what they call FWAS). However, the authors need to validate their findings. I have two major concerns:

1) The authors use PredictDB to impute organ-specific transcript abundances and use these for model generation. PredictDB is from a separate publication (in 2015) and I do not know how the data generated are validated, but the authors should at least validate their findings here by e.g. using data from biopsies where there is both genome sequence data and transcript data.

We thank the reviewer for their complementary words on our work. The PredictDB models are the standard for the genomic field, having been widely used and assessed, e.g. (Gamazon et al. 2015) with >700 citations. In their original Gamazon et al. publication, the models were validated both through cross-validation and in an external set of *GTEX* samples not used in model training (Gamazon et al. 2015). PredictDB models for specific organs have also been validated in other publications (Li et al. 2018; Tavares et al. 2021; Hale et al. 2021). Further validation of the PredictDB models would be beyond the scope of our reaction flux work. For the convenience of the reader, we have expanded the description of PredictDB to reflect these validations:

*The elastic net models from PredictDB (Gamazon et al. 2015) were used to impute organ-specific gene expression levels from individual-level genotypes. These are well-established models that have been extensively validated (Gamazon et al. 2015; Li et al. 2018; Tavares et al. 2021; Hale et al. 2021). The latest release of PredictDB models, which had been trained with *GTEX* v8 data, were obtained from <https://predictdb.org/>. They were used with *PLINK2* (Chang et al. 2015) to predict relative transcripts abundances using genotype data from the *INTERVAL* (Moore et al. 2014; Di Angelantonio et al. 2017) and *UK Biobank* (Sudlow et al. 2015) cohorts.*

2) The authors say they are using the Recon family of GEMs. What does this mean? Which of the models? Why not use the far superior models presented more recently, i.e. Human1, which has been shown to have much less gaps and errors, and be much better in predicting fluxes. The latter becomes in particular important as the authors aim to calculate fluxes.

We thank the reviewer for this excellent suggestion concerning the use of HUMAN1. In the initially submitted manuscript, we had used organ-specific models extracted from the Harvey/Harvetta multiorgan models(Thiele et al. 2020). We chose to use these models because Thiele et al. had done extensive research to build a set of curated organ-specific networks of all the major organs of the human body, including physiological organ-specific boundaries for the exchange reactions between blood and each organ. Harvey/Harvetta had been built using the RECON3D human reconstruction as a template.

Following the reviewer's advice, we have now performed extensive re-analysis using HUMAN1 based-models. We achieve this by performing a liftover (i.e., transferring) Harvey/Harvetta models to HUMAN1. This allows us to leverage both the extensive curation of the Harvey/Harvetta models and the improved annotations of HUMAN1. The liftover process is based on the fact that 97% of HUMAN1 reactions can be mapped to equivalent Recon3D reactions and it has been detailed in the Methods subsection "**Building organ-specific models**". It is worth noting that as part of the liftover process, the new HUMAN1-based models inherit the boundaries for the exchange reactions of Harvey/Harvetta; Hence the new models are not unbounded.

We recomputed genetically personalized flux values with the new models and repeated the FWAS to blood metabolome and CAD risk. We find that a significant core of flux-phenotype associations was detected with both HUMAN1 and RECON3D models. However, as expected, we also find novel associations, as well as associations from the original models that were not reproduced in the HUMAN1 models. These differences come from the improved annotations of HUMAN1, which can have a significant impact on the simulated flux values and, as the reviewer states, HUMAN1 models are expected to be able to achieve better flux predictions than Recon3D-based models(Robinson et al. 2020).

We now focus the manuscript on the results obtained with the HUMAN1 models, which has required significantly adaptation of the Results section to reflect the new associations. We also dedicate a paragraph to discussing the differences between Recon3D and HUMAN1 results:

Finally, we also evaluated the effect of the underlying genome-scale reconstructions of human metabolism in the FWAS for blood metabolic features. With this aim, we used organ-specific models built from the Recon3D human GSMM(Brunk et al. 2018; Thiele et al. 2020) to compute genetically-personalised fluxes for the INTERVAL cohort(Moore et al. 2014; Di Angelantonio et al. 2017), test their association to blood metabolic features, and compare the results to the above-described FWAS that had used fluxes computed with HUMAN1-based models. We identified 3,895 significant associations between blood metabolic features and the genetically personalized flux values computed using Recon3D-based organ-specific models (Table S2). There was a significant overlap with HUMAN1 models as 1,761 of these associations could be replicated in the HUMAN1-based FWAS, and the associated effect sizes on blood metabolites were highly correlated between HUMAN1 and Recon3D analyses($\rho=0.72$). However, 2,134 associations were only statistically significant on the Recon3D-based analysis and could not be replicated with HUMAN1 models. Likewise, of the 4,312 significant associations between blood metabolic features and fluxes computed using HUMAN1 models, 2,551 associations could not be detected with Recon3D-based models. Such discrepancy between HUMAN1- and Recon3D-based analyses is not surprising; HUMAN1(Robinson et al. 2020), which is a newer reconstruction of human metabolism than Recon3D(Brunk et al. 2018), expands gene reaction annotations and refines reaction reversibility, both of which can have significant effects on how genetic variation propagates through the network

and, thus, can lead to significant differences in the resulting personalized flux maps and the downstream FWAS. Indeed, many discrepancies between the Recon3D and HUMAN1 results are likely artifacts emerging from erroneous or incomplete annotations in Recon3D. Throughout this work, we focus on the analyses and discussion of HUMAN1-based fluxes, as HUMAN1 has been established to be a better representation of human metabolism (Robinson et al. 2020), but results obtained with Recon3D-based models will also be provided in the appropriate supplementary tables (Table S2, Table S3, and Table S4).

Most importantly, the reviewer-suggested shift to HUMAN1 has resulted in a significantly improved analysis while also preserving our main findings and conclusions.

This also leads to another criticism, namely why not use the GECKO approach for modeling. This has been shown again to significantly improve flux predictions. In fact I think the resolution of the authors findings could be significantly improved by doing this. I am aware that this is basically asking the authors to redo a lot of their analysis, but it is not good for the research community that old models that have been shown to be poor for flux simulations are used in new and exciting studies like the one presented here.

We thank the reviewer for their suggestion of the use of GECKO; however, given that we do not have relevant proteomics data, it is unlikely to significantly improve the simulated fluxes.

Firstly, we agree that GECKO can be a genuinely transformative algorithm when proteomics measures are available, and HUMAN1 is uniquely suited to use the algorithm by providing catalytic constants for the enzymes in the network (Robinson et al. 2020). Unfortunately, we only have access to transcriptomics for our analyses. We were unable to find any database that could provide us with absolute proteomics measures for the modelled organs as well as models that could allow us to impute personalised organ-specific protein levels for them.

We are, of course, aware that GECKO can also be applied without proteomics measures (the protein pool approach). Briefly, as the reviewer may know, this approach works by having each enzyme-catalysed reaction consume a proportion of the total protein pool, enabling to restrict the hypothetical maximum flux through reactions in the network (Robinson et al. 2020). This reduces the unboundedness of GSMs, which, if not addressed can lead to unrealistically high flux values. However, we must note that our organ-specific models are not unbounded, as we have physiological organ-specific constraints in the exchange rates in each organ, and this limits the potential benefits of GECKO. Indeed, to quote the HUMAN1 (Robinson et al. 2020) paper where the protein pool approach is presented and evaluated:

The greatest advantages and improvement in flux predictions are achieved when experimental exchange rates are limited or unavailable, which is most often the case when modeling human systems. However, when such flux measurements are available, the potential improvement offered by enzyme constraints becomes limited, as illustrated in the most constrained simulation in Fig. 5D

Additionally, in our workflow, we also make use of the GIM3E algorithm (Schmidt et al. 2013). The algorithm applies reaction flux minimization weighted by gene expression to constraint reaction fluxes. This allows calculating flux distributions that are consistent with enzyme transcript levels in each organ and restricting the maximum flux through reactions. The latter achieves a similar effect to GECKO (protein pool) in constraining maximum fluxes, but GIM3E also enables to leverage transcriptomics data to compute flux maps consistent with enzyme transcript levels, a capability not presently supported by GECKO.

In summary, we argue that given that our models have physiological boundaries for the exchange rates of metabolites and that we use the GIM3E algorithm to leverage transcriptomics data and

constraint maximum reaction flux values, using GECKO(with protein pool) would not offer a significant improvement on the simulated flux values. We note, however, that this is only because we do not have access to absolute proteomics measures; if that were the case, we agree that using GECKO would significantly improve the accuracy of the simulated flux values. We have added the following sentences in the discussion to reflect this:

Our analysis has several limitations.(...) Additionally, while transcript levels are widely used in genome-scale metabolic modelling(Schmidt et al. 2013; Jamialahmadi et al. 2019; Robinson et al. 2020), protein levels have a more direct effect on enzymatic activity, and new methods are being developed to fully integrate them into GSMs(Robinson et al. 2020; Sánchez et al. 2017). With models to impute the levels of proteins becoming increasingly available(Xu et al. 2022; Wingo et al. 2021), we expect that the framework for computing genetically personalized fluxes will be extended to integrate the protein layer in the future.

References

- Agren, Rasmus, Adil Mardinoglu, Anna Asplund, Caroline Kampf, Mathias Uhlen, and Jens Nielsen. 2014. "Identification of Anticancer Drugs for Hepatocellular Carcinoma through Personalized Genome-Scale Metabolic Modeling." *Molecular Systems Biology* 10 (3): 721–721. <https://doi.org/10.1002/msb.145122>.
- Angelantonio, Emanuele Di, Simon G. Thompson, Stephen Kaptoge, Carmel Moore, Matthew Walker, Jane Armitage, Willem H. Ouwehand, David J. Roberts, John Danesh, and INTERVAL Trial Group. 2017. "Efficiency and Safety of Varying the Frequency of Whole Blood Donation (INTERVAL): A Randomised Trial of 45 000 Donors." *Lancet (London, England)* 390 (10110): 2360–71. [https://doi.org/10.1016/S0140-6736\(17\)31928-1](https://doi.org/10.1016/S0140-6736(17)31928-1).
- Bershad, S, A Rubinstein, J R Paterniti, N A Le, S C Poliak, B Heller, H N Ginsberg, R Fleischmajer, and W V Brown. 1985. "Changes in Plasma Lipids and Lipoproteins during Isotretinoin Therapy for Acne." *The New England Journal of Medicine* 313 (16): 981–85. <https://doi.org/10.1056/NEJM198510173131604>.
- Brunk, Elizabeth, Swagatika Sahoo, Daniel C. Zielinski, Ali Altunkaya, Andreas Dräger, Nathan Mih, Francesco Gatto, et al. 2018. "Recon3D Enables a Three-Dimensional View of Gene Variation in Human Metabolism." *Nature Biotechnology* 36 (3): 272–81. <https://doi.org/10.1038/nbt.4072>.
- Chang, Christopher C, Carson C Chow, Laurent Cam Tellier, Shashaank Vattikuti, Shaun M Purcell, and James J Lee. 2015. "Second-Generation PLINK: Rising to the Challenge of Larger and Richer Datasets." *GigaScience* 4: 7. <https://doi.org/10.1186/s13742-015-0047-8>.
- Dickinson, Jared M, and Blake B Rasmussen. 2013. "Amino Acid Transporters in the Regulation of Human Skeletal Muscle Protein Metabolism." *Current Opinion in Clinical Nutrition and Metabolic Care* 16 (6): 638–44. <https://doi.org/10.1097/MCO.0b013e3283653ec5>.
- Drake, Kenneth J, Veniamin Y Sidorov, Owen P McGuinness, David H Wasserman, and John P Wiksw. 2012. "Amino Acids as Metabolic Substrates during Cardiac Ischemia." *Experimental Biology and Medicine (Maywood, N.J.)* 237 (12): 1369–78. <https://doi.org/10.1258/ebm.2012.012025>.
- Folger, Ori, Livnat Jerby, Christian Frezza, Eyal Gottlieb, Eytan Ruppim, and Tomer Shlomi. 2014. "Predicting Selective Drug Targets in Cancer through Metabolic Networks." *Molecular Systems Biology* 7 (501): 501–501. <https://doi.org/10.1038/msb.2011.35>.
- Gallagher, D., S. Chung, and M. Akram. 2013. "Body Composition." In *Encyclopedia of Human Nutrition*, 191–99. Elsevier. <https://doi.org/10.1016/B978-0-12-375083-9.00027-1>.
- Gamazon, Eric R, Heather E Wheeler, Kaanan P Shah, Sahar V Mozaffari, Keston Aquino-Michaels, Robert J Carroll, Anne E Eyler, et al. 2015. "A Gene-Based Association Method for Mapping Traits Using Reference Transcriptome Data." *Nature Genetics* 47 (9): 1091–98. <https://doi.org/10.1038/ng.3367>.
- GTEX Consortium. 2013. "The Genotype-Tissue Expression (GTEx) Project." *Nature Genetics* 45 (6): 580–85. <https://doi.org/10.1038/ng.2653>.
- Hale, Andrew T, Lisa Bastarache, Diego M Morales, John C. Wellons, David D Limbrick, and Eric R Gamazon. 2021. "Multi-Omic Analysis Elucidates the Genetic Basis of Hydrocephalus." *Cell Reports* 35 (5): 109085. <https://doi.org/10.1016/j.celrep.2021.109085>.
- Jamialahmadi, Oveis, Sameereh Hashemi-Najafabadi, Ehsan Motamedian, Stefano Romeo, and Fatemeh Bagheri. 2019. "A Benchmark-Driven Approach to Reconstruct Metabolic Networks for Studying Cancer Metabolism." *PLoS Computational Biology* 15 (4): e1006936. <https://doi.org/10.1371/journal.pcbi.1006936>.
- Jamshidi, Neema, and Bernhard Palsson. 2006. "Systems Biology of SNPs." *Molecular Systems Biology* 2: 1–4. <https://doi.org/10.1038/msb4100077>.

- Julkunen, Heli, Anna Cichońska, P. Eline Slagboom, and Peter Würtz. 2021. "Metabolic Biomarker Profiling for Identification of Susceptibility to Severe Pneumonia and COVID-19 in the General Population." *ELife* 10: 1–20. <https://doi.org/10.7554/eLife.63033>.
- Li, Binglan, Shefali S Verma, Yogasudha C Veturi, Anurag Verma, Yuki Bradford, David W Haas, and Marylyn D Ritchie. 2018. "Evaluation of PrediXcan for Prioritizing GWAS Associations and Predicting Gene Expression." *Pacific Symposium on Biocomputing. Pacific Symposium on Biocomputing* 23 (10): 448–59. <https://doi.org/10.3390/genes12101531>.
- Moore, Carmel, Jennifer Sambrook, Matthew Walker, Zoe Tolkien, Stephen Kaptoge, David Allen, Susan Mehenny, et al. 2014. "The INTERVAL Trial to Determine Whether Intervals between Blood Donations Can Be Safely and Acceptably Decreased to Optimise Blood Supply: Study Protocol for a Randomised Controlled Trial." *Trials* 15 (1): 363. <https://doi.org/10.1186/1745-6215-15-363>.
- Murray, J C, R S Gilgor, and G S Lazarus. 1983. "Serum Triglyceride Elevation Following High-Dose Vitamin A Treatment for Pityriasis Rubra Pilaris." *Archives of Dermatology* 119 (8): 675–76. <http://www.ncbi.nlm.nih.gov/pubmed/6870321>.
- Raškevičius, Vytautas, Valeryia Mikalayeva, Ieva Antanavičiūtė, Ieva Ceslevičienė, Vytenis Arvydas Skeberdis, Visvaldas Kairys, and Sergio Bordel. 2018. "Genome Scale Metabolic Models as Tools for Drug Design and Personalized Medicine." *PLoS ONE* 13 (1): 1–14. <https://doi.org/10.1371/journal.pone.0190636>.
- Redlich, C. A., J. S. Chung, M. R. Cullen, W. S. Blaner, A. M. Van Bennekum, and L. Berglund. 1999. "Effect of Long-Term Beta-Carotene and Vitamin A on Serum Cholesterol and Triglyceride Levels among Participants in the Carotene and Retinol Efficacy Trial (CARET)." *Atherosclerosis* 145 (2): 425–32. [https://doi.org/10.1016/S0021-9150\(99\)00266-X](https://doi.org/10.1016/S0021-9150(99)00266-X).
- Ritchie, Scott C, Praveen Surendran, Savita Karthikeyan, Samuel A Lambert, Thomas Bolton, Lisa Pennells, John Danesh, Emanuele Di Angelantonio, Adam S Butterworth, and Michael Inouye. 2021. "Quality Control and Removal of Technical Variation of NMR Metabolic Biomarker Data in ~120,000 UK Biobank Participants." *MedRxiv* 9: 2021.09.24.21264079. <https://www.medrxiv.org/content/10.1101/2021.09.24.21264079v1%0Ahttps://www.medrxiv.org/content/10.1101/2021.09.24.21264079v1.abstract>.
- Robinson, Jonathan L., Pinar Kocabaş, Hao Wang, Pierre-Etienne Cholley, Daniel Cook, Avlant Nilsson, Mihail Anton, et al. 2020. "An Atlas of Human Metabolism." *Science Signaling* 13 (624): 1–12. <https://doi.org/10.1126/scisignal.aaz1482>.
- Sánchez, Benjamín J, Cheng Zhang, Avlant Nilsson, Petri-Jaan Lahtvee, Eduard J Kerkhoven, and Jens Nielsen. 2017. "Improving the Phenotype Predictions of a Yeast Genome-Scale Metabolic Model by Incorporating Enzymatic Constraints." *Molecular Systems Biology* 13 (8): 935. <https://doi.org/10.15252/msb.20167411>.
- Schmidt, Brian J, Ali Ebrahim, Thomas O Metz, Joshua N Adkins, Bernhard Ø Palsson, and Daniel R Hyduke. 2013. "GIM3E: Condition-Specific Models of Cellular Metabolism Developed from Metabolomics and Expression Data." *Bioinformatics (Oxford, England)* 29 (22): 2900–2908. <https://doi.org/10.1093/bioinformatics/btt493>.
- Sudlow, Cathie, John Gallacher, Naomi Allen, Valerie Beral, Paul Burton, John Danesh, Paul Downey, et al. 2015. "UK Biobank: An Open Access Resource for Identifying the Causes of a Wide Range of Complex Diseases of Middle and Old Age." *PLoS Medicine* 12 (3): 1–10. <https://doi.org/10.1371/journal.pmed.1001779>.
- Tavares, Vânia, Joana Monteiro, Evangelos Vassos, Jonathan Coleman, and Diana Prata. 2021. "Evaluation of Genotype-Based Gene Expression Model Performance: A Cross-Framework and Cross-Dataset Study." *Genes* 12 (10). <https://doi.org/10.3390/genes12101531>.
- Thiele, Ines, Swagatika Sahoo, Almut Heinken, Johannes Hertel, Laurent Heirendt, Maike K

- Aurich, and Ronan MT Fleming. 2020. "Personalized Whole-body Models Integrate Metabolism, Physiology, and the Gut Microbiome." *Molecular Systems Biology*. <https://doi.org/10.15252/msb.20198982>.
- Vahlquist, C, G Michaëlsson, A Vahlquist, and B Vessby. 1985. "A Sequential Comparison of Etretinate (Tigason) and Isotretinoin (Roaccutane) with Special Regard to Their Effects on Serum Lipoproteins." *The British Journal of Dermatology* 112 (1): 69–76. <https://doi.org/10.1111/j.1365-2133.1985.tb02293.x>.
- Wingo, Thomas S, Yue Liu, Ekaterina S. Gerasimov, Jake Gockley, Benjamin A. Logsdon, Duc M. Duong, Eric B. Dammer, et al. 2021. "Brain Proteome-Wide Association Study Implicates Novel Proteins in Depression Pathogenesis." *Nature Neuroscience* 24 (6): 810–17. <https://doi.org/10.1038/s41593-021-00832-6>.
- Xu, Yu, Scott C Ritchie, Yujian Liang, Paul R. H. J. Timmers, Maik Pietzner, Loic Lannelongue, Samuel A. Lambert, et al. 2022. "An Atlas of Genetic Scores to Predict Multi-Omic Traits." *BioRxiv* 15 (Mi): 2022.04.17.488593. <https://www.biorxiv.org/content/10.1101/2022.04.17.488593v1%0Ahttps://www.biorxiv.org/content/10.1101/2022.04.17.488593v1.abstract>.
- Yizhak, Keren, Orshay Gabay, Haim Cohen, and Eytan Rupp. 2013. "Model-Based Identification of Drug Targets That Revert Disrupted Metabolism and Its Application to Ageing." *Nature Communications* 4 (1): 2632. <https://doi.org/10.1038/ncomms3632>.

REVIEWERS' COMMENTS

Reviewer #1 (Remarks to the Author):

The authors have improved the revised manuscript by including detailed explanations of the methods and performing additional analysis. The codes are available for reproducibility.

Reviewer #2 (Remarks to the Author):

I think the authors have addressed all reviewer comments very well and I do not have any further comments.